# Chemogenomics for NR1 nuclear hormone receptors

Laura Isigkeit [1,4], Espen Schallmayer[1,4], Romy Busch[2], Lorene Brunello [1,3], Amelie Menge [1,3], Lewis Elson[1,3], Susanne Müller [1,3], Stefan Knapp [1,3], Alexandra Stolz [3], Julian A. Marschner[2] & Daniel Merk [1,2] ✉

Nuclear receptors (NRs) regulate transcription in response to ligand binding and NR modulation allows pharmacological control of gene expression. Although some NRs are relevant as drug targets, the NR1 family, which comprises 19 NRs binding to hormones, vitamins, and lipid metabolites, has only been partially explored from a translational perspective. To enable systematic target identification and validation for this protein family in phenotypic settings, we present an NR1 chemogenomic (CG) compound set optimized for complementary activity/selectivity profiles and chemical diversity. Based on broad profiling of candidates for specificity, toxicity, and off-target liabilities, sixty-nine comprehensively annotated NR1 agonists, antagonists and inverse agonists covering all members of the NR1 family and meeting potency and selectivity standards are included in the final NR1 CG set. Proof-of-concept application of this set reveals effects of NR1 members in autophagy, neuroinflammation and cancer cell death, and confirms the suitability of the set for target identification and validation.

The discovery of new chemical modalities, mediating desired therapeutic effects is an essential step in the development of novel medicines. At the interface of chemical biology and phenotypic screening, chemogenomics (CG) enables the identification and validation of new therapeutic targets as a prerequisite for rational drug discovery[1]. In CG, libraries of highly annotated biologically active compounds are screened for (desired) phenotypic outcomes in disease-relevant models[1,2]. In contrast to chemical probes which are considered the highest quality chemical tools for such purpose, molecules forming a CG set may exhibit less stringent potency and selectivity properties and are assembled considering broader selectivity profiles but ideally non-overlapping off-target activity that allows deconvolution of the underlying mechanisms of an observed phenotypic outcome[1,3]. Thereby, CG enables target identification and validation in the absence of chemical probes for most protein targets[4]. CG libraries have been compiled and used as tool for target identification, for example, to elucidate the roles and therapeutic potential of kinases (e.g., KCGS[5]),

or with a focus on cardiovascular targets[6], highlighting the potential of CG in early drug discovery. However, although medicinal chemistry in academia and industry has developed potent chemical tools for hundreds of macromolecular targets, their rational assembly to broadly annotated CG sets with sufficient chemical diversity and non-overlapping selectivity profiles is poorly developed and often not available to the public[1,2,4].

Here, we describe the compilation and preliminary application of a CG library for the nuclear receptor (NR) family NR1 for which no such dedicated set is currently available to the best of our knowledge. As ligand-sensing transcription factors, the 48 human NR transcriptionally regulate innumerable physiological processes and represent attractive drug targets[7]. The NR1 family comprises 19 NRs and, together with the NR3 family of steroid hormone receptors, is the best-studied part of NRs[8]. It is subdivided into seven subfamilies (NR1A, NR1B, NR1C, NR1D, NR1F, NR1H, NR1I) based on phylogenetic relationship (Fig. 1)[9,10]. Among the NR1 receptors are targets of

[1]Goethe University Frankfurt, Institute of Pharmaceutical Chemistry, Frankfurt, Germany. [2]Ludwig-Maximilians-Universität (LMU) München, Department of Pharmacy, Munich, Germany. [3]Buchmann Institute for Molecular Life Sciences and Institute of Biochemistry 2, Goethe University Frankfurt, Frankfurt, Germany. [4]These authors contributed equally: Laura Isigkeit, Espen Schallmayer. ✉e-mail: daniel.merk@cup.lmu.de

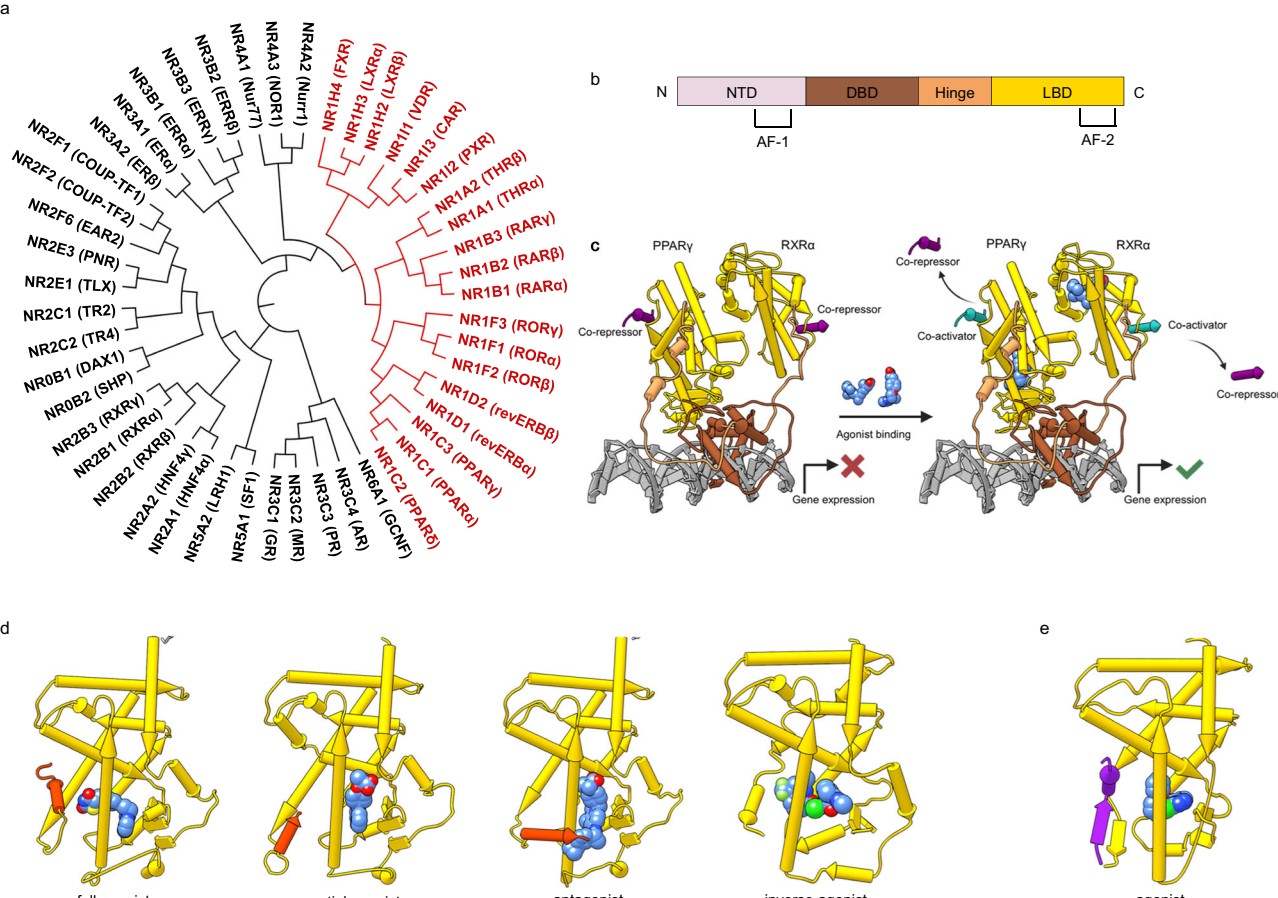

**Fig. 1 | Structure and function of NR1 receptors. a** Phylogenetic tree of the NR family comprising 48 members in humans. NR1 family in red. **b** The archetypal domain structure of NRs is composed of an unordered N-terminal domain (NTD), containing the ligand-independent activation function 1 (AF-1), a DNA binding domain (DBD) comprising two zinc finger motifs, a flexible hinge region, and a ligand binding domain (LBD) with the ligand-dependent activation function AF-2. **c** Molecular mechanism of NR activity. The example shows the full-length PPARγ-RXRα heterodimer (pdb id: 3dzy[68], colors corresponding to modular domain structure in (**b**)) bound to DNA (gray). In their inactive conformation, in the absence of a ligand, NRs bind co-repressors (dark purple), resulting in repression of gene expression. Agonist binding (blue) induces the active conformation, leading to co-repressor displacement and co-activator (cyan, pdb id: 2fvj[69] for co-activator placement) recruitment to activate gene expression. **d** NR modulation by different

types of ligands involves different conformational changes in the NR LBD affecting the position of the AF-2 (red): agonists stabilize an active conformation with AF-2 bound to the LBD core (PPARγ with bound agonist rosiglitazone; pdb id: 7awc[70]); partial agonists cause weaker stabilization with potentially shifted AF-2 (PPARγ with bound partial agonist AL26-29; pdb id: 5hzc[71]); antagonists prevent agonist binding and do not stabilize the active conformation (PPARγ with bound antagonist SR11023; pdb id: 6c5t[72]); inverse agonists block the constitutive activity of NRs like RORs by stabilizing the inactive state (RORγ with bound inverse agonist; pdb id: 6slz). Structural and molecular mechanisms of NR modulation have been reviewed in[7,73,74]. **e** Some NRs like NR1D act primarily as transcriptional repressors and recruit co-repressors (revERBα with agonist and co-repressor NCoR1 in purple; pdb id: 8d8i[75]).

important drugs such as thyroid hormone receptors (THR, NR1A) and peroxisome proliferator-activated receptors (PPAR, NR1C) but also significantly less studied receptors such as revERB (NR1D)[8]. In addition to validated therapeutic roles, NR1 receptors hold great promise in multiple pathologies, including, for example, immunity and inflammation, neurodegeneration, and cancer[11–14]. CG may offer an avenue to capture the elusive potential of NR1 receptors, but a well-characterized CG library to interrogate the biology of this protein family is lacking despite the availability of many NR1 ligands.

To close this gap, we here identify suitable NR1 modulators in the literature to compile a CG set and optimize their combination for on-target potency, complementary selectivity profiles, and chemical diversity to ensure orthogonality. Extensive profiling for nonspecific effects, activity on critical off-targets, and in-family selectivity support the selection to form a CG set. Cellular applications of this library reveal the involvement of NR1 receptors in neuroinflammation, autophagy, and cancer cell death, highlighting the NR1 CG set as a

valuable tool to explore additional roles of this important protein family.

## Results

### Selection of CG compound candidates for NR1

Our initial selection of CG compound candidates was based on compound-bioactivity annotations available in public repositories (PubChem[15], ChEMBL[16], IUPHAR/BPS[17], BindingDB[18] and Probes&Drugs[19]) which we have previously compiled in a curated dataset[20]. NR1 CG compound candidates were identified among the annotated NR1 ligands (30862 compounds with potency ≤10 μM) in this dataset based on i) community agreed criteria[21,22] for cellular potency (≤10 μM, preferably ≤1 μM) and selectivity (up to five off-targets at final concentration), ii) chemical diversity, iii) diverse modes of action (agonist, antagonist and inverse agonist), and iv) commercial availability using KNIME[23] for extraction. Although prioritizing commercially available compounds may result in the omission of some potentially more potent NR1 modulators reported in the literature, full

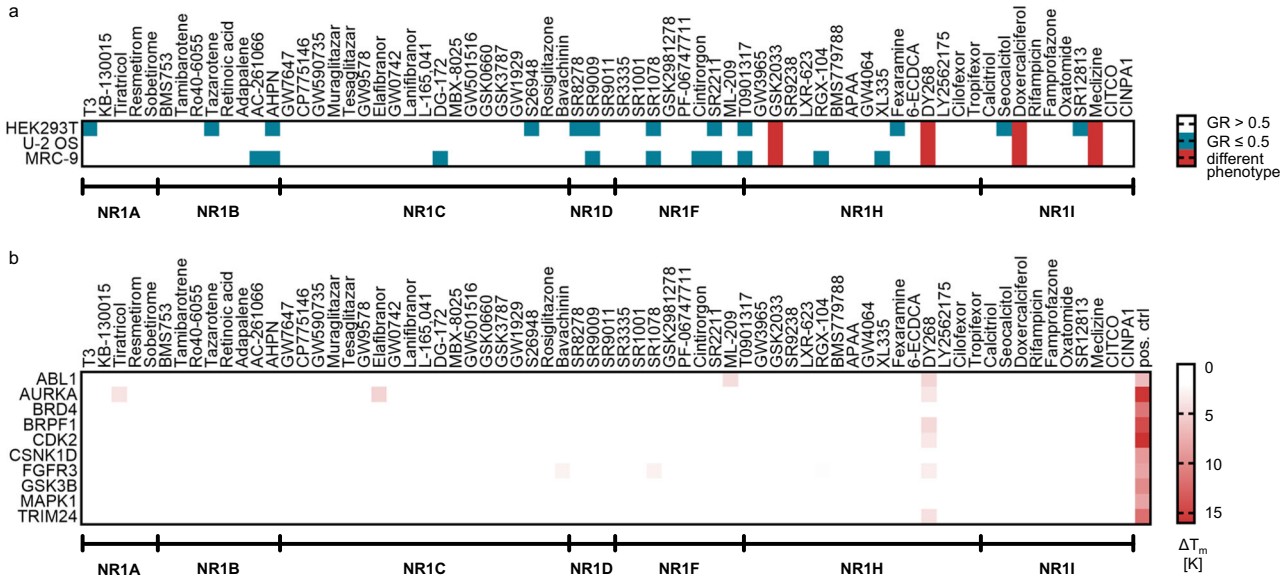

**Fig. 2 | Toxicity and off-target liability profiling of the NR1 CG set. a** CG compound candidate (10 μM) profiling for cytotoxic (GR < 0.5) or phenotypic effects on three cell lines (HEK293T, U-2 OS and MRC-9) after 24 h incubation[55]; n = 2. **b** CG compound candidate (20 μM) profiling for off-target binding in a liability target screening by differential scanning fluorimetry. Proteins were used at 2 μM; staurosporine (ABL1, AURKA, CDK2, FGFR3 and GSK3B), (+)-JQ1 (BRD4), GSK6853 (BRPF1), PK016714a (CSNK1D), GDC-0994 (MAPK1) and IACS-9571 (TRIM24) served as positive controls (pos. ctrl) at a concentration of 20 μM. The heatmap shows the mean ΔTm calculated by the Boltzmann fit; n = 2.

commercial availability of the CG set was desirable to enable access and use by a broad community. For NR1 receptors with multiple available modulators matching the criteria, the chemical diversity of the ligands was analyzed based on the Tanimoto similarity[24] (Morgan fingerprints (radius=2)[25]) of the compounds and their Murcko molecular frameworks (skeletons)[26] and served as selection criterion. We obtained an initial set of 80 chemically diverse CG compound candidates for the NR1 family (Supplementary Table 1) covering all NR1 subfamilies and complying with our criteria according to the literature.

**Activity profiling and CG compound selection**

The CG compound candidates were acquired and analyzed for identity and purity (≥95%) by NMR, LC-UV, LC-ELSD, and LC-MS (analytical data available for reference in the CG compound sheets and at https://doi.org/10.5281/zenodo.10474037). After this initial quality control, all candidates were profiled in vitro to assess their suitability for inclusion in the NR1 CG set. Compounds were triaged by first assessing them in a primary cell viability assay[27] in three cell lines (HEK293T, U-2 OS and MRC-9 fibroblasts; Fig. 2a). This assay relies on confluence measurement by microscopy at different time points (6 h, 12 h, 18 h and 24 h) to determine the growth rate (GR), indicative of growth inhibiting (GR < 1) or cytotoxic (GR < 0) effects. Most CG compound candidates for NR1 showed no effect on cell viability at 10 μM. Nine compounds (AHPN, SR9009, SR1078, SR2211, T0901317, GSK2033, DY268, doxercalciferol, and meclizine) exhibiting a GR ≤ 0.5 or inducing atypical cellular phenotypes upon visual inspection were further evaluated in a high-content microscopy-based multiplex assay[28,29] in the same cell lines over time (12 h, 24 h and 48 h compound treatment) to capture phenotypic features characteristic for cell health effects (apoptosis, alterations in cytoskeleton, membrane permeabilization, mitochondrial mass) using orthogonal stains (Supplementary Fig. 1). The assay also identifies precipitated, non-soluble compounds at the concentrations used. In this secondary toxicity profiling, DY268 caused a moderate decrease in HEK293T and U-2 OS but not MRC-9 cell numbers indicating a mild toxicity which appeared to be mediated by effects on tubulin according to the multiplex results. At a concentration of 1 μM, this effect was considered weak enough for further

evaluation and inclusion in the set. Nevertheless, a possible moderate cytotoxicity should be taken into account when analyzing the results of DY268 in CG applications. None of the other CG compound candidates assessed showed any phenotypic effects in the multiplex assay or had any effect on the cell count over time up to 48 h at up to 10 μM.

To annotate potential activity on critical off-targets, the CG compound candidates were additionally screened by differential scanning fluorimetry (DSF) for binding to a panel of representative kinases and bromodomains that are highly ligandable and/or cause strong phenotypic outcomes when inhibited[30–32]. The bromodomain containing proteins BRD4, TRIM24 and BRPF1 represent three diverse bromodomain subfamilies and are the most ligandable proteins within this protein family. BRD4 inhibition additionally elicits a strong phenotype in many cellular systems[33]. The evaluated kinases are likewise representative of different subfamilies, highly ligandable and exhibit critical cellular roles: AURKA representing a typical AGC kinase, CDK2 representing CDKs (CGMK), and MAPK1 (ERK) representing MAPKs are highly ligandable; GSK3B/CSNK1D are kinases representing a specific binding mode (back pocket binder); ABL1 represents a typical soluble tyrosine kinase and FGFR3 a receptor tyrosine kinase. Additionally, AURKA, CDK2 and GSK3B elicit strong phenotypes when inhibited[34–36]. Based on the maximum variance of the liability DSF assay, a compound-induced increase in the protein melting temperature (ΔTm) > 1.8 °C (≥ 2 × SD) was considered relevant and we observed this effect for seven compounds (tiratricol, elafibranor, bavachinin, SR1078, ML-209, RGX-104 and DY268) on at least one liability target (Fig. 2b). DY268 (20 μM) exhibited weak interaction with several liability targets which might lead to phenotypic effects as also observed in the multiplex assay, which are not related to the main NR target. However, no effect on the liability targets was observed for DY268 at 1 μM (Supplementary Fig. 2), suggesting that this concentration is suitable for CG applications.

In-family selectivity profiling (Fig. 3a, Supplementary Fig. 3) of the CG compound candidates was performed in uniform hybrid reporter gene assays[37] on the main target and on all NRs in the respective subfamily. A few compounds (KB-130015, resmetirom, DG-172, GSK0660, SR8278, SR9009, SR9011, SR3335, ML-209, GW3965,

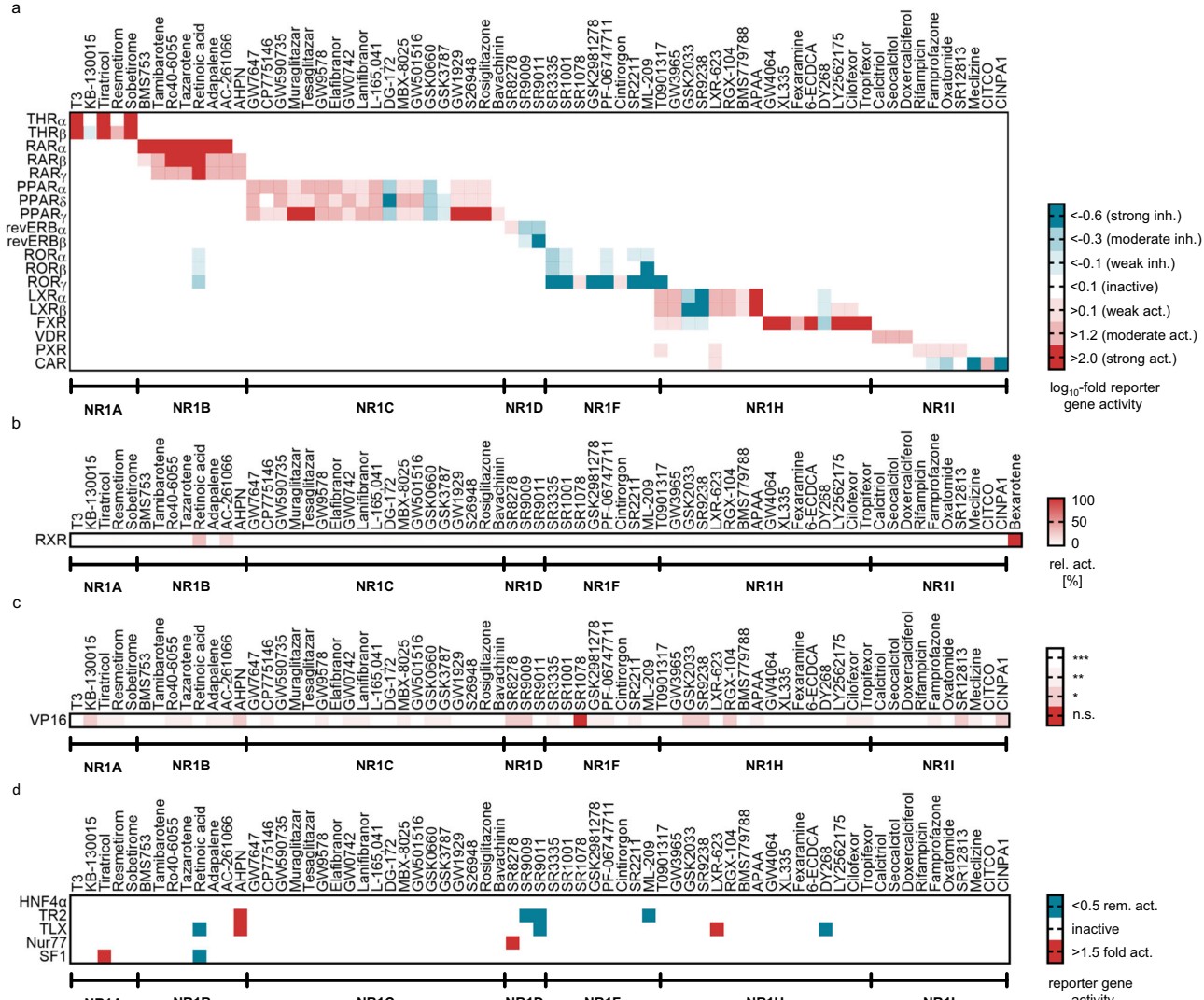

**Fig. 3 | Selectivity screening of the NR1 CG set. a** In-family selectivity profiles of NR1 CG compounds at the recommended concentrations (cf. Figure 4). The heatmap shows NR mediated activation (red; agonists) and inhibition of reporter gene expression (blue; antagonists and inverse agonists), expressed as mean $\log_{10}$ fold reporter activity. Since revERBs (NR1D) act as transcriptional repressors, inverse agonists cause activation and agonists cause inhibition of gene expression; all activities within a compound's target subfamily were determined in uniform hybrid reporter gene assays; $n = 3$; activities outside the subfamilies reported in literature (retinoic acid[76–78], GW7647[79], SR9009[80], LXR-623[81]) were checked and only confirmed for retinoic acid (NR1F) and LXR-623 (NR1I) at the recommended concentrations (cf. Figure 4). **b** Effects of NR1 CG compounds on RXRα activity (NR2B1). Heatmap shows the mean relative RXR activation by the CG compounds at the recommended concentrations compared to reference agonist bexarotene (1 μM); $n = 3$. **c** Effects of the NR1 CG compounds on the ligand-independent transcriptional inducer Gal4-VP16[43,44] to capture non-specific effects on transcriptional activity. The heatmap shows the significance level of effects on the main NR1 target vs. effects on VP16 activity (n.s. – not significant ($p \geq 0.05$), * $p < 0.05$, ** $p < 0.01$, *** $p < 0.001$; two-sided t-test); $n = 3$. **d** Selectivity profiling of NR1 CG compounds at the recommended concentrations on representative NRs outside the NR1 family. The heatmap shows NR mediated activation (red; agonists) and inhibition of reporter gene expression (blue; antagonists and inverse agonists), expressed as mean reporter activity relative to DMSO control. Since TR2 (NR2C1) and TLX (NR2E1) act as transcriptional repressors, inverse agonists cause activation and agonists cause inhibition of reporter gene expression. HNF4α (NR2A1), Nur77 (NR4A1) and SF1 (NR5A1) exhibit constitutive activity. Activities were determined in uniform hybrid reporter gene assays; $n = 3$.

GSK2033, LXR-623, RGX-104, rifampicin, oxatomide, and CINPA1) weakly modulated their main target at 1 μM but exhibited the intended activity at a compound concentration of 3 μM or 10 μM, respectively. Eight compounds (oleoyl ethanolamide, LG101506, XY018, IMB-808, GSK4112, 25(R)–27-hydroxycholesterol, pleconaril and clotrimazole) were excluded after the in-family profiling as their activities reported in the literature were not reproduced even at higher concentration (up to 30 μM). Amiodarone (THR antagonist) and lovastatin (PXR agonist) exhibited the intended activity on their respective NR target but were excluded for activities outside the NR family (Supplementary Table 2, Supplementary Fig. 4)[38–42] that might cause strong phenotypes and since several better suitable CG compounds for THR and PXR were

identified. The remaining CG compound candidates were further profiled for RXR activation (Fig. 3b) to exclude heterodimer-mediated activity and for non-specific effects on reporter activity using the ligand-independent transcriptional inducer Gal4-VP16[43,44] (Fig. 3c). Apart from retinoic acid and the synthetic retinoid AC-261006, all compounds were found to be selective over RXR. The putative CAR antagonist PK11195 was excluded due to its non-specific effects on multiple NRs and in the Gal4-VP16 assay (Supplementary Fig. 4), while the weak activity of SR1078 on Gal4-VP16 induced reporter activity was considered acceptable. To capture the selectivity of the CG compound candidates on NRs outside the NR1 family, we tested their activity on representative NR2 (HNF4α, NR2A1; TR2, NR2C1; TLX, NR2E1), NR4

| Target family | Compound | Main NR target | p(Potency) | Type | NR off-target | Recom. conc. |
|---|---|---|---|---|---|---|
| NR1A | T3 | 1A1; 1A2 | 8.7; 8.7 | Agonist | - | 1 µM |
| | KB-130015 | 1A2 | 5.7 | Antagonist | - | 3 µM |
| | Tiratricol | 1A1; 1A2 | 8.7; 8.4 | Agonist | - | 1 µM |
| | Resmetirom | 1A2 | 6.7 | Agonist | - | 10 µM |
| | Sobetirome | 1A1; 1A2 | 7.3; 8.2 | Agonist | - | 1 µM |
| NR1B | BMS753 | 1B1 | 7.0 | Agonist | - | 1 µM |
| | Tamibarotene | 1B1; 1B2; 1B3 | 7.3; 6.7; 6.2 | Agonist | - | 1 µM |
| | Ro40-6055 | 1B1; 1B2; 1B3 | 9.5; 8.0; 8.0 | Agonist | - | 1 µM |
| | Tazarotene | 1B1; 1B2; 1B3 | 7.2; 9.1; 7.4 | Agonist | - | 1 µM |
| | Retinoic acid | 1B1; 1B2; 1B3 | 6.4; 6.4; 6.5 | Agonist | 1F1/F2/F3; 2B1 | 1 µM |
| | Adapalene | 1B1; 1B2; 1B3 | 7.7; 8.7; 8.5 | Agonist | - | 1 µM |
| | AC-261066 | 1B1; 1B2; 1B3 | 6.2; 8.0; 6.3 | Agonist | 2B1 | 1 µM |
| | AHPN | 1B2; 1B3 | 6.7; 7.3 | Agonist | - | 1 µM |
| NR1C | GW7647 | 1C1; 1C3 | 8.2; 6.0 | Agonist | 1C2 | 1 µM |
| | CP775146 | 1C1 | 7.2 | Agonist | - | 1 µM |
| | GW590735 | 1C1; 1C2 | 8.4; 6.8 | Agonist | - | 1 µM |
| | Muraglitazar | 1C1; 1C3 | 7.3; 7.8 | Agonist | 1C2 | 1 µM |
| | Tesaglitazar | 1C1; 1C3 | 6.5; 6.9 | Agonist | - | 1 µM |
| | GW9578 | 1C1; 1C3 | 7.3; 6.0 | Agonist | 1C2 | 1 µM |
| | Elafibranor | 1C1; 1C2; 1C3 | 6.9; 6.7; 6.4 | Agonist | - | 1 µM |
| | GW0742 | 1C2 | 9.0 | Agonist | 1C1/C3 | 1 µM |
| | Lanifibranor | 1C2; 1C3 | 6.0; 6.7 | Agonist | 1C1 | 1 µM |
| | L-165,041 | 1C1; 1C2; 1C3 | 6.0; 7.8; 6.0 | Agonist | - | 1 µM |
| | DG-172 | 1C2 | 5.7 | Antagonist | - | 10 µM |
| | MBX-8025 | 1C2 | 8.7 | Agonist | - | 1 µM |
| | GW501516 | 1C2 | 8.9 | Agonist | - | 1 µM |
| | GSK0660 | 1C1; 1C2; 1C3 | 5.2; 5.3; 5.1 | Antagonist | - | 10 µM |
| | GSK3787 | 1C2 | 7.0 | Antagonist | 1C3 | 1 µM |
| | GW1929 | 1C3 | 8.2 | Agonist | - | 1 µM |
| | S26948 | 1C3 | 7.4 | Agonist | - | 1 µM |
| | Rosiglitazone | 1C3 | 7.5 | Agonist | - | 1 µM |
| | Bavachinin | 1C3 | 6.2 | Agonist | - | 1 µM |
| NR1D | SR8278 | 1D1 | 6.3 | Antagonist | - | 10 µM |
| | SR9009 | 1D1; 1D2 | 5.2; 5.0 | Agonist | - | 10 µM |
| | SR9011 | 1D1; 1D2 | 5.7; 5.5 | Agonist | - | 10 µM |

| Target family | Compound | Main NR target | p(Potency) | Type | NR off-target | Recom. conc. |
|---|---|---|---|---|---|---|
| NR1F | SR3335 | 1F1; 1F2; 1F3 | 6.3; 5.8; 5.5 | inv. Agonist | - | 10 µM |
| | SR1001 | 1F1; 1F2; 1F3 | 6.0; 6.0; 7.0 | inv. Agonist | - | 1 µM |
| | SR1078 | 1F3 | 5.7 | Agonist | - | 1 µM |
| | GSK2981278 | 1F3 | 7.7 | inv. Agonist | - | 1 µM |
| | PF-06747711 | 1F3 | 8.4 | inv. Agonist | - | 1 µM |
| | Cintirorgon | 1F3 | 7.7 | Agonist | - | 1 µM |
| | SR2211 | 1F3 | 6.5 | inv. Agonist | - | 1 µM |
| | ML-209[a] | 1F2; 1F3 | 5.9; 6.7 | inv. Agonist | 1F1 | 10; 1 µM |
| | T0901317 | 1F3 | 6.3 | inv. Agonist | 1H4 | 1 µM |
| NR1H | GW3965 | 1H2; 1H3 | 5.7; 6.7 | Agonist | - | 3 µM |
| | GSK2033 | 1H2; 1H3 | 7.4; 7.0 | Antagonist | - | 3 µM |
| | SR9238 | 1H2; 1H3 | 7.4; 6.7 | Antagonist | - | 1 µM |
| | LXR-623 | 1H2; 1H3 | 5.4; 5.2 | Agonist | 1I2 | 3 µM |
| | RGX-104 | 1H2; 1H3 | 5.5; 5.5 | Agonist | - | 10 µM |
| | BMS779788 | 1H2; 1H3 | 6.5; 6.7 | Agonist | - | 1 µM |
| | APAA | 1H2; 1H3 | 9.0; 9.0 | Agonist | - | 1 µM |
| | GW4064 | 1H4 | 6.4 | Agonist | - | 1 µM |
| | T0901317 | 1H2; 1H3 | 6.7; 6.4 | Agonist | 1H4 | 1 µM |
| | XL335 | 1H4 | 7.0 | Agonist | - | 1 µM |
| | Fexaramine | 1H4 | 6.0 | Agonist | - | 1 µM |
| | 6-ECDCA | 1H4 | 6.5 | Agonist | - | 1 µM |
| | DY268 | 1H4 | 6.2 | Antagonist | 1H2/H3 | 1 µM |
| | LY2562175 | 1H4 | 6.7 | Agonist | - | 1 µM |
| | Cilofexor | 1H4 | 7.4 | Agonist | - | 1 µM |
| | Tropifexor | 1H4 | 9.5 | Agonist | - | 1 µM |
| NR1I | Calcitriol | 1I1 | 7.0 | Agonist | - | 1 µM |
| | Seocalcitol | 1I1 | 7.0 | Agonist | - | 1 µM |
| | Doxercaliferol | 1I1 | 6.0 | Agonist | - | 1 µM |
| | Rifampicin | 1I2 | 6.0 | Agonist | - | 10 µM |
| | Famprofazone | 1I2 | 6.9 | Agonist | - | 1 µM |
| | Oxatomide | 1I2 | 5.2 | Agonist | - | 10 µM |
| | SR12813 | 1I2 | 6.9 | Agonist | - | 1 µM |
| | T0901317 | 1I2 | 7.4 | Agonist | 1H4 | 1 µM |
| | Meclizine | 1I3 | 7.2 | inv. Agonist | - | 1 µM |
| | CITCO | 1I3 | 7.5 | Agonist | - | 1 µM |
| | CINPA1 | 1I3 | 6.2 | Antagonist | - | 10 µM |

**Fig. 4 | Final set of 69 NR1 CG compounds.** Main targets, potency, activity type, relevant NR off-targets and recommended concentrations of the final 69 NR1 CG compounds. p(Potency) values refer to the pEC$_{50}$ for agonists and to pIC$_{50}$ for antagonists/inverse agonists. Colors denote different NR1 subfamilies and match with the colors in Fig. 5. [a] ML-209 is recommended for CG at two different concentrations: 1 µM is selective for NR1F3 (RORγ); 10 µM inhibits both NR1F2 (RORβ) and NR1F3 (RORγ) and has weak inhibitory effects on NR1F1 (RORα; NR off-target).

(Nur77, NR4A1) and NR5 (SF1, NR5A1) receptors which generally revealed high selectivity for the intended NR1 targets (Fig. 3d). A few off-target activities were detected for individual compounds but no set of CG compounds for an NR1 subfamily shared a common off-target. Since CG evaluates consistent effects of a compound set, such isolated off-targets of individual compounds are tolerable, and the NR selectivity results, therefore, corroborated all remaining NR1 CG candidates.

The comprehensive in vitro profiling for activity and selectivity confirmed 69 compounds (Fig. 4, Fig. 5) as suitable to form a CG set covering the NR1 family. Despite not all being fully selective for a single

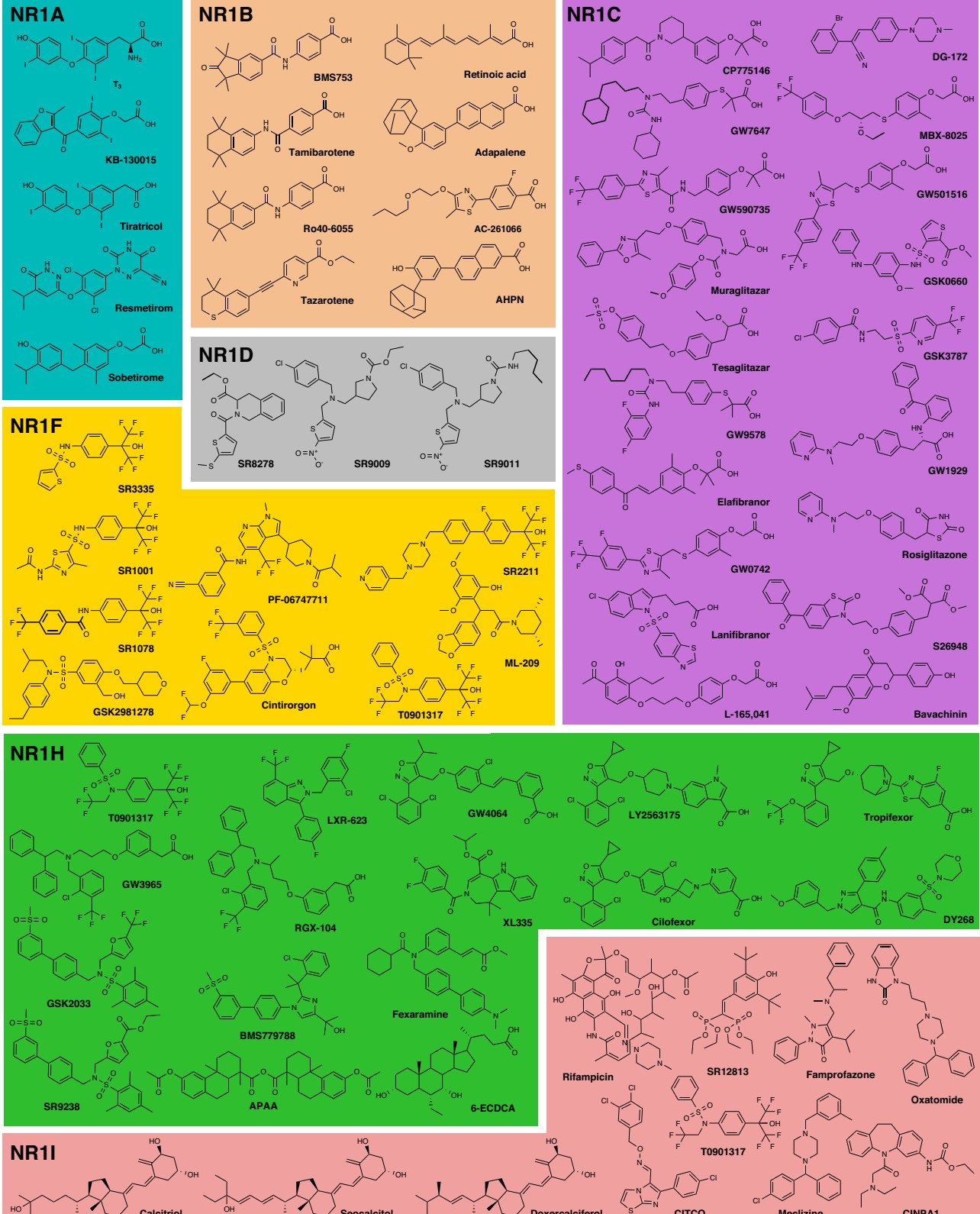

**Fig. 5 | Structures of the final set of NR1 CG compounds.** Shown are the structures of the selected CG compounds as well as the most commonly used names in the literature. Colors denote different NR1 subfamilies and match with the colors in Fig. 4.

NR, these selected compounds exhibited non-overlapping target profiles, thus allowing target deconvolution when used as a CG set. Based on the results of potency, selectivity, and toxicity profiling, suitable concentrations for CG application were identified for all compounds in the set (Fig. 4). At these recommended concentrations for cell-based studies, the selected compounds robustly modulate their intended NR target, exhibit favorable selectivity, and lack relevant cytotoxicity as well as critical off-target effects in the liability panel.

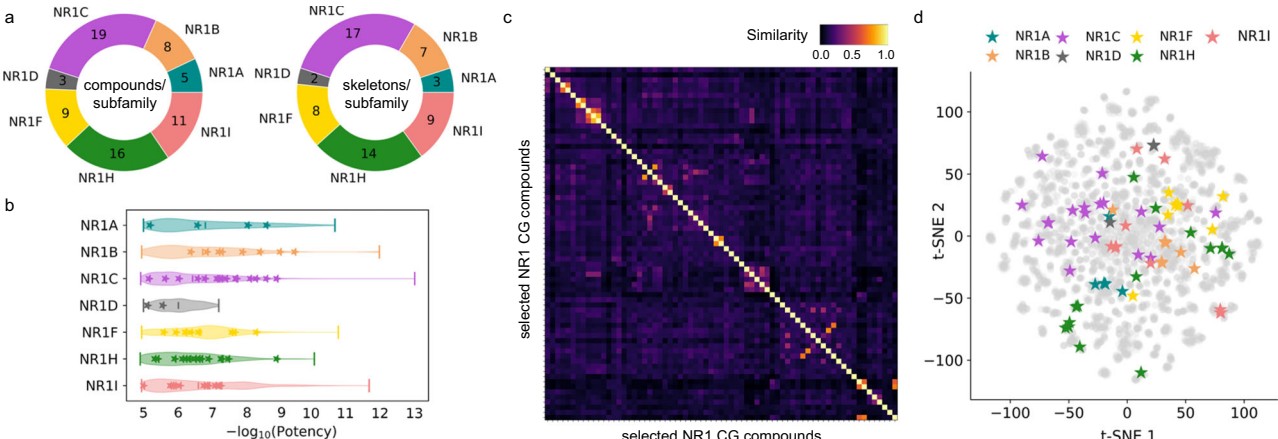

**Fig. 6 | Feature analysis of the NR1 CG compounds. a** Pie charts of the number of compounds per subfamily and the number of diverse skeletons per subfamily. **b** Potency distribution of the CG compounds is shown as the negative decadic logarithm of potency. Violin plots represent the potency distribution of all known ligands of the respective subfamily (≤ 100 μM from dataset[20]) and stars represent the selected CG compounds. **c** Pairwise Jaccard-Tanimoto similarity heatmap of the selected NR1 CG compounds computed on Morgan fingerprints. **d** t-SNE plot of known NR1 ligands (gray, ≤ 100 μM from dataset[20]) with the selected NR1 CG compounds highlighted and colored by their NR target subfamilies.

## Characteristics and chemical features of the NR1 CG set

The 69 CG compounds emerging from the comprehensive profiling broadly covered the NR1 family with five compounds for NR1A, eight for NR1B, 19 for NR1C, three for NR1D, nine for NR1F, 16 for NR1H, and eleven for NR1I (Fig. 6a) and exhibited a potency ≤1 μM with few exceptions (Fig. 6b): Weaker potency had to be accepted, for example, to cover the NR1D subfamily and to increase chemical diversity we prioritized KB-130015 (NR1A antagonist), oxatomide (NR1I agonist), and bavachinin (NR1C agonist) despite potency > 1 μM. Selecting chemically diverse ligands was particularly challenging for the NR1A and NR1D subfamilies where only few ligand chemotypes were available. Still, computational evaluation of the final CG set revealed low pairwise Tanimoto similarity[24] computed on Morgan fingerprints with very few exceptions (Fig. 6c) underscoring high chemical diversity of the selected NR1 CG compounds. This was also evident from scaffold analysis (Supplementary Fig. 5) which demonstrated that the 69 selected CG compounds represented 57 different chemotypes. Chemical space visualization by t-Distributed Stochastic Neighbor Embedding[45] (t-SNE; Fig. 6d), which is more effective than principal component analysis (PCA) in preserving the local structure of data and can capture non-linear relationships between features, further highlighted broad distribution of the selected compounds over the chemical space of known NR1 modulators and illustrated the chemical diversity of the ligands within and across subfamilies. Adding chemical orthogonality for robust application to CG-based target identification and validation studies is an important feature of the NR1 CG set as it reduces the probability for common elusive off-targets[46,47]. Additionally, the set incorporates all mechanistic types of NR ligands (agonists, antagonists and inverse agonists) for further improving confident target identification and validation in phenotypic screening. Although the majority of NR1 receptors display no relevant constitutive activity in the absence of ligands, the presence of NR1A, NR1C, NR1D, NR1H and NR1I antagonists in the CG set may reveal phenotypic effects of blocking the activity of natural ligands present in cellular settings. For the constitutively active NR1F receptors, inverse agonists have been prioritized in the CG set but are likewise accompanied by agonists for orthogonality.

The identification of ligands with non-overlapping selectivity profiles provided another challenge since ligands exhibiting full selectivity within NR subfamilies are rare, *e.g.*, in the NR1C and NR1I subfamilies. Nevertheless, compounds with diverse non-overlapping selectivity profiles were available to assemble the NR1 CG set in a manner enabling deconvolution of effects to the NR1 targets (Fig. 3a). It should be noted, however, that subtype selectivity has not been established within all NR1 subfamilies and while sufficiently selective ligands are available for the individual NR1A, NR1C, NR1F, NR1H, and NR1I receptors, full deconvolution to a single receptor may not be possible for the NR1B and NR1D receptors due to the lack of subtype-selective ligands.

## Applications of the NR1 CG set for target identification

With the final NR1 CG set in hand we probed its application in diverse phenotypic settings to evaluate the suitability of the set for linking phenotypes to specific targets and to potentially discover additional pharmacologies for NR1. Following the concept of CG, the library was used as a set at a single concentration (see Fig. 4) to reveal effects in a target-centric fashion.

Based on the ability of several NRs to counteract nuclear factor κB (NF-κB) activity and inflammation and on preliminary evidence suggesting that some NRs mediate anti-neuroinflammatory effects[12], we studied the impact of the NR1 CG set on NF-κB activity in astrocytes (T98G, Fig. 7a). Decreased NF-κB activity was consistently observed for RAR (NR1B) agonists with the exception of the only RARα selective agonist BMS753, indicating an anti-inflammatory role of RARβ and/or RARγ in this cell type. Similarly, non-selective inverse ROR (NR1F) agonists (SR3335, SR1001, T0901317) counteracted NF-κB activity, while the RORγ selective ligands (PF-06747711, cintirorgon) had less effects. The NR1D family (revERB) emerged as another potential target to achieve anti-neuroinflammatory effects as the revERB antagonist SR8278 enhanced and the agonists SR9009 and SR9011 diminished NF-κB activity. Moreover, vitamin D receptor (VDR, NR1I1) agonists coherently enhanced and PXR agonists slightly diminished NF-κB activity while no consistent effects were detected for NR1A, NR1C and NR1H ligands.

Next, we interrogated the potential role of NR1 receptors in autophagy, a critical cellular mechanism to maintain homeostasis[48]. Autophagy can enable the cells' survival through starvation but is also involved in the removal of pathogens and misfolded proteins, and its dysregulation has been associated with various pathologies like cancer, neurodegeneration and metabolic dysfunction[48,49]. We evaluated the effects of the NR1 CG set on autophagy over time, using an autophagic flux assay in RPE1 cells harboring an LC3 reporter (GFP-LC3-RPF-LC3ΔG)[50] (Fig. 7b). RAR (NR1B) agonists, apart from the RARα selective BMS753, markedly and consistently increased autophagic

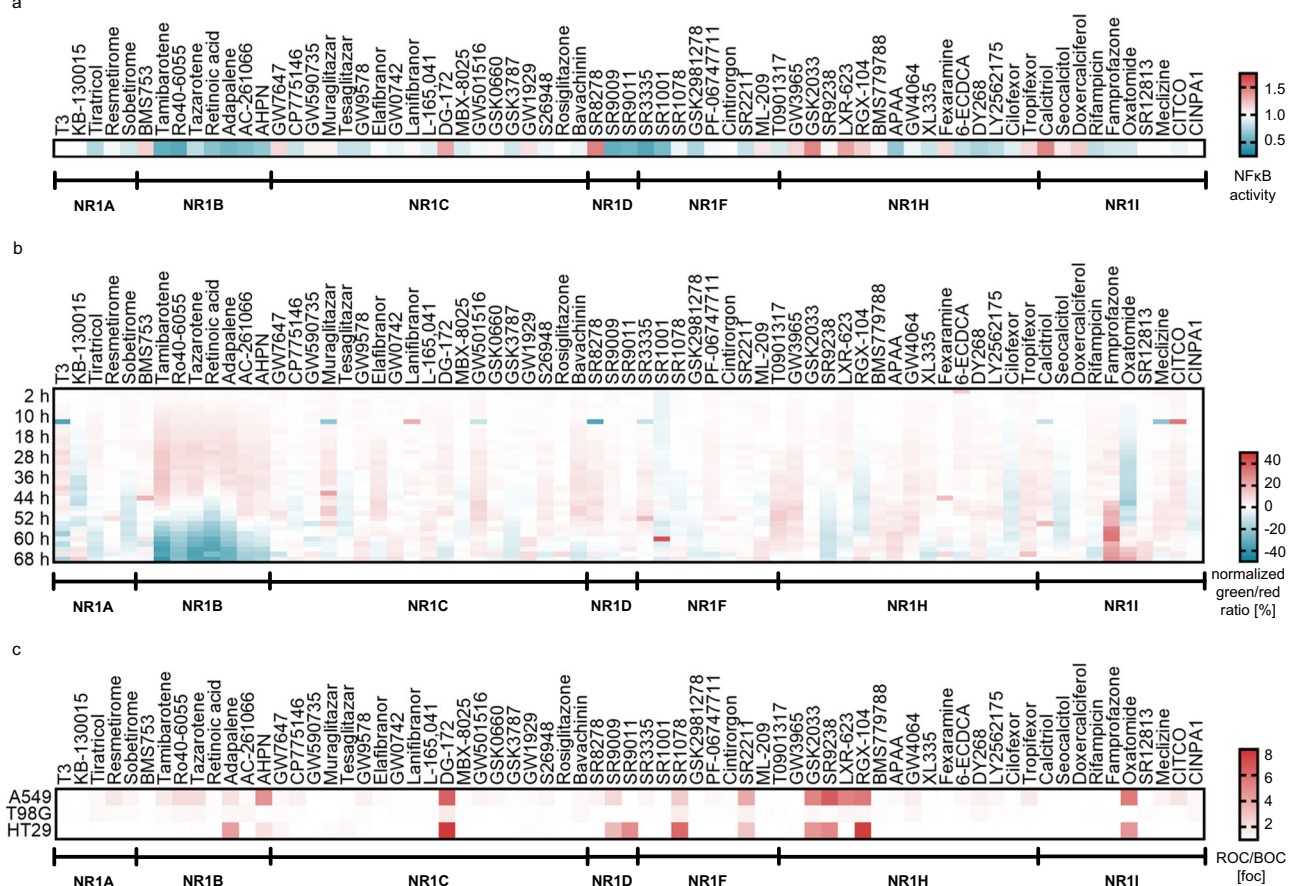

**Fig. 7 | Applications of the NR1 CG set to target identification in vitro.** (a) Effects of the NR1 CG set on NFκB activity in astrocytes (T98G). The heatmap shows mean NF-κB activity compared to DMSO (0.1%) treated cells; n = 4. (b) Effects of the NR1 CG set on autophagic flux over time (hours after treatment) in RPE1 GFP-LC3-RPF-LC3ΔG cells[50]. The heatmap shows the mean normalized green/red fluorescence ratio compared to DMSO (0.1%) treated cells; n = 3. A lower green/red fluorescence ratio indicates increased autophagic flux. (c) Induction of cell death by the NR1 CG set in A549, T98G and HT29 cells expressed as necrotic cell counts (=ROC, red fluorescent dye detecting permeable cell membranes) per total cell counts (=BOC, blue fluorescent dye detecting cell nuclei). The heatmap shows the red/blue object count ratio normalized to DMSO (0.1%) treated cells; n = 3.

flux in a delayed fashion, indicating a genomic effect. These results, therefore, suggest a role of RAR (NR1B) in autophagy and a potential of RAR ligands for pharmacological modulation of this crucial process. Interestingly, effects of the natural RAR agonist retinoic acid on autophagy have been reported previously[51] but were referred to mechanisms involving other pathways than RAR activation.

Phenotypic effects of NR modulation were also evident in the application of the NR1 CG set to a cancer proliferation (Supplementary Fig. 6) and cell death assay (Fig. 7c). Liver X receptor (NR1H2/3) agonists inhibited proliferation of A549 lung cancer cells and HT29 colorectal adenocarcinoma cells and markedly enhanced cell death in these cells. Interestingly, T98G cells were not responsive, suggesting cell-specific effects of LXR activation. Additionally, cancer cell proliferation was diminished by treatment with revERB (NR1D) agonists but not by the revERB antagonist SR8278.

## Discussion

Despite remarkable drug development efforts for some NR1 members, the therapeutic relevance of this protein family is still limited to a few targets and indications, such as PPARγ agonists in type 2 diabetes and RAR agonists in cancer. However, the central regulatory roles of NR1 receptors in physiological homeostasis suggest much more therapeutic potential for these proteins beyond the few existing indications that remain unexplored. By synergizing the strengths of complex phenotypic models and highly annotated sets of chemical tools, CG

may reveal the uncharted potential of NR1 receptors. We have, therefore, assembled a CG set for NR1 to aid and promote target identification and validation studies on this important protein family. Multiple ligand chemotypes are available for most NR1 receptors from extensive medicinal chemistry programs, but many of these bioactive compounds lack full specificity, requiring an appropriate combination of chemically diverse NR1 ligands with non-overlapping activity profiles from this vast collection. Additionally, not all reported activities were confirmed in our assays, further highlighting the need for chemical tool validation to prevent false hypotheses from phenotypic results observed with inappropriate tools[52].

Based on extensive validation of the intended on-target activities in a uniform assay platform for all 19 NR1 receptors, as well as broad profiling for selectivity, toxicity, and non-specific effects, 69 compounds were deliberately selected to form the final NR1 CG library. This set was additionally optimized for chemical diversity with multiple orthogonal chemical tools per NR1 protein, and all selected NR1 ligands are commercially available, rendering the set sustainable and broadly accessible. These characteristics make the NR1 CG set a valuable and validated tool to explore the uncharted biology of the NR1 family. Extensive characterization data and recommendations are available for all evaluated NR1 ligands (source data file and CG compound sheets in the Supplementary Data, and at www.eubopen.org) as a resource for the application of the set and to deconvolute phenotypic effects to molecular modes of action.

Consistent phenotypic effects of CG compound subsets for individual NR1 subfamilies suggested the involvement of NR1 receptors in autophagy, neuroinflammation and cancer cell death which may be further explored. These proof-of-concept applications demonstrated the suitability and potential of the set to link biological readouts to targets in complex cellular settings at medium throughput. The fully annotated, validated and broadly available NR1 CG set can thus advance research on this important protein family and reveal unprecedented therapeutic opportunities.

# Methods

## CG compound selection

**Computational methods for CG candidate compound selection.** The initial compound selection was performed with the Software Konstanz Information Miner (KNIME, version 4.5)[23] with RDKit (version 2022.09.1) nodes. Compounds were extracted from a recently compiled dataset[20] containing multiple annotated bioactivities from five public databases (PubChem[15] ChEMBL[16], IUPHAR/BPS[17], BindingDB[18] and Probes&Drugs[19]). Compounds were prefiltered for bioactivity on the intended main NR target based on the annotated bioactivities. The number of off-targets was determined by searching the dataset for other targets of the preselected compounds with annotated bioactivities at ≤10 μM. Commercial availability as another selection criterion was determined by combining vendor databases (Tocris, Sigma, Cayman Chemicals, Selleckchem, and MilliporeSigma) with the compound selection using ChEMBL IDs and SMILES. The chemical diversity of the CG candidate compounds was evaluated based on Morgan fingerprints[25] which were calculated using the RDKit Fingerprint node with the following settings: Fingerprint type = Morgan, number of bits = 1024 and radius = 2. Skeletons were extracted using RDKit Find Murcko Scaffolds with the setting create frameworks. Up to 10 chemically diverse molecules and skeletons were selected for each target using the RDKit Diversity Picker. Bioactivity values annotated in the database were compared/validated with the original literature before candidates were selected for profiling. Property analysis was done using KNIME and python (version 3.8)[53]. Basic descriptors (molweight, cLogP) were calculated using RDKit Descriptor Calculation. For the t-SNE and potency analysis, all NR1 ligands with annotated bioactivity ≤100 μM were selected from the dataset. The t-SNE was computed based on Morgan fingerprints (from RDKit in KNIME) using scikit-learn (version 1.3.)[54] and the following settings: n_components = 2, perplexity = 30, learning_rate = auto, init = pca. The similarity heatmap was generated using Tanimoto similarity computed on Morgan fingerprints.

## Chemical quality control

**General QC by LC-UV and LC-MS.** All CG compounds were subjected to initial chemical quality control by LC-MS using an Agilent 1260 Infinity II HPLC (Agilent, Waldbronn, Germany, consisting of a 1260 Infinity II Flexible Pump (#G7104C), a 1260 Infinity II Multisampler (#G67167A), a 1260 Infinity II Multicolumn Thermostat (#G67116A) and a 1260 Infinity II Diode Array Detector HS (DAD, G7117C) coupled to an Agilent single quadrupole MS detector (G6125B, ESI pos. 100–1000)). Chromatography was performed using an InfinityLab Poroshell 120 Bonus-RP, 2.1 × 100 mm, 2.7 μm column (Agilent #695768-901 T) coupled to an Infinity Lab Poroshell 120 Bonus-RP, 2.1 × 5 mm, 2.7 μm, UHPLC guard (Agilent #821725-925). As mobile phase water + 0.1% formic acid (solvent A) and acetonitrile + 0.1% formic acid (solvent B) were used. Method A: 7.50 min at a flow rate of 0.600 mL/min starting with 95% solvent A for 0.4 min followed by a linear gradient to 100% solvent B after 6.3 min. Method B: 7.50 min at a flow rate of 0.600 mL/min starting with 95% solvent A for 0.5 min followed by a linear gradient to 100% solvent B after 5.2 min. Method C: 7.50 min at a flow rate of 0.600 mL/min with the following gradient: 0 min: 5% B − 2 min: 80% B − 5 min: 95% B − 7 min: 95% B. For purity analysis, absorbance was measured at five wavelengths in the range of 245 − 395 nm using the DAD. For identity analysis, M + H$^+$ was monitored in the TIC (100–1000 Da) and in SIM mode. Compounds for which no M + H$^+$ was detected in the initial QC and compounds with poor UV absorbance were subjected to further analysis by LC-MS/MS and/or LC-ELSD.

**QC by LC-ESI-MS/MS.** Compound identity was analyzed by LC-ESI-MS/MS using an API 5500 QTRAP triple quadrupole mass spectrometer with a TurboV-ion source (Sciex, Darmstadt, Germany) coupled to an Agilent 1260 HPLC system (Agilent, Waldbronn, Germany) and a SIL-20A/HT autosampler (Shimadzu, Duisburg, Germany) under control of Analyst 1.6 software (Sciex). A Zorbax SBAq (3.5 μm, 100 mm × 3 mm, Agilent, protected with a 0.5 μm and a 0.2 μm frit) was used as stationary phase in combination with 0.1% formic acid (A) and acetonitrile (B) as mobile phase at a flow rate of 500 μL/min. Compounds were investigated by 5 μL injection of 100 nM sample solutions (10 mM stock solutions in DMSO diluted in A/B: 50/50, v/v) under isocratic conditions as indicated for the individual compounds. MS detection was performed under positive or negative ESI conditions recording significant mass transitions for corresponding parent ions in the MRM mode (see CG compound sheets in the Supplementary Data). Compound purity was analyzed using an Agilent 1100 HPLC system (Agilent, Waldbronn, Germany, consisting of a #G1311A pump, a #G1316A thermostated column compartment and a DAD #G1315B) with a SIL-20A/HT autosampler (Shimadzu, Duisburg, Germany) coupled to an API 3200 QTRAP triple quadrupole mass spectrometer with a TurboV-ion source (Sciex, Darmstadt, Germany) under control of Analyst 1.6 software (Sciex). A Zorbax SBAq (3.5 μm, 100 mm × 3 mm, Agilent, protected with a 0.5 μm and a 0.2 μm frit) was used as stationary phase in combination with 0.1% formic acid (A) and acetonitrile (B) as mobile phase at a flow rate of 500 μL/min. Compounds were investigated by 10 μL injections of 10 μM sample solutions (10 mM stock solutions in DMSO diluted in the mobile phase components A and B, unless otherwise specified) under isocratic conditions as indicated for the individual compounds (see CG compound sheets in the Supplementary Data). UV detection was performed at 5 wavelengths in the range 210−325 nm (see CG compound sheets in the Supplementary Data). The TWC and one meaningful XWC per compound are reported in the CG compound sheets (Supplementary Data). For purity analysis by UV, MS data were recorded to confirm the identity of the analyzed signal but were not shown. For purity analysis by MS, the mass spectrometer was operated in the Q1 scan mode under ESI-positive conditions with otherwise identical conditions.

**QC by LC-ELSD.** Analysis by LC-ELSD was performed with a Shimadzu HPLC system (Shimadzu, Duisburg, Germany, consisting of a LC-40D pump, an SPD-M40 DAD, an RF-20A XS fluorescence detector an ELSD LT II and a CBM-40 system controller) controlled by LabSolutions 5.87 software (Shimadzu). A Zorbax SBAq (3.5 μm, 100 mm × 3 mm, Agilent, protected with a 0.5 μm and a 0.2 μm frit) was used as stationary phase in combination with 0.1% formic acid (A) and acetonitrile (B) as mobile phase at a flow rate of 500 μL/min. The following ELSD settings were used: T: 40 °C, N$_2$: 350 kPa, gain: 7. Compounds were investigated by 20 μL injections of 10 μM or 100 μM sample solutions (10 mM stock solutions in DMSO or acetonitrile diluted in the mobile phase components A and B) under isocratic conditions as indicated for the individual compounds (CG compound sheets in the Supplementary Data).

**NMR.** $^1$H and $^{13}$C NMR spectra were recorded at 25 °C on Bruker Avance III HD 400, or Avance III HD 500 spectrometers equipped with a CryoProbeTM Prodigy broadband probe (Bruker Corporation, Billerica, MA, USA). The NMR spectra were calibrated using the proton or carbon signals of residual nondeuterated solvent peaks (2.50 and 39.52 ppm for DMSO-$d_6$; 5.32 and 53.84 ppm for CD$_2$Cl$_2$). NMR spectra were

processed and analyzed with MNova (version 12.0.1-20560 Mestrelab Research, Santiago de Compostela, A Coruña, Spain). NMR spectra are provided in the CG compound sheets (Supplementary Data).

## Biological quality control

**Incucyte viability assay.** HEK293T (ATCC, CRL-1573) and U-2 OS (ATCC, HTB-96) cells were cultured in DMEM high glucose supplemented with 10% fetal bovine serum (FBS), penicillin (100 U/mL), and streptomycin (100 µg/mL), MRC-9 cells (ATCC, CCL-212) were cultured in EMEM supplemented with 10% FBS, penicillin (100 U/mL), and streptomycin (100 µg/mL) at 37 °C and 5% $CO_2$. HEK293T, U-2 OS (1500 cells/well) and MRC-9 (1250 cells/well) cells were seeded into 384 well plates in a final volume of 50 µL. Cells were cultured overnight before blank images were taken using the Incucyte® S3 Live-Cell Analysis (Sartorius AG, Goettingen, Germany, #4647) instrument. Test compounds and reference controls were added using the Echo 550 liquid handler to final concentrations of 10 µM and each plate was imaged over a course of 24 h with images being taken at 6 h increments. Experiments were run as biological duplicates. Confluence data of all time points was analyzed using Incucyte® Base Analysis software (Sartorius AG) for calculation of the growth rate (GR)[55] to determine compound effects. Compounds deemed as hits (growth rate ≤0.5 on at least two different cell lines or with atypical phenotype) were further evaluated in the multiplex assay.

**Multiplex High-Content assay.** For hit evaluation after the primary viability assessment, a live cell high-content assay[28,29] was performed in HEK293T (ATCC, CRL-1573), U-2 OS (ATCC, HTB-96) and MRC-9 (ATCC, CCL-212) cells. HEK293T and U-2 OS cells were cultured in DMEM high glucose supplemented with 10% FBS, penicillin (100 U/mL), and streptomycin (100 µg/mL), MRC-9 cells were cultured in EMEM supplemented with 10% FBS, penicillin (100 U/mL), and streptomycin (100 µg/mL) at 37 °C and 5% $CO_2$. HEK293T, U-2 OS (1250 cells/well) and MRC-9 (1750 cells/well) cells were seeded into 384 well plates and stained simultaneously, with 60 nM Hoechst 33342 (Thermo Scientific, #62249), 75 nM MitoTracker™ Deep Red FM (Invitrogen, #M22426), 0.3 µl/well Annexin V Alexa Fluor® 680 conjugate (Invitrogen, #A35109) and 25 nL/well BioTracker™ 488 Green Microtubule Cytoskeleton Dye (Sigma Aldrich, #SCT142). Fluorescence and cellular shape were measured before compound treatment and after 12 h, 24 h and 48 h compound exposure, respectively using the CQ1 high-content confocal microscope (Yokogawa, Ratingen, Germany). Experiments were performed in biological duplicates. The following parameters were used for image acquisition: Ex 405 nm/Em 447/60 nm, 500 ms, 50%; Ex 561 nm/Em 617/73 nm, 100 ms, 40%; Ex 488/Em 525/50 nm, 50 ms, 40%; bright field, 300 ms, 100% transmission, one centered field per well, 7 z stacks per well with 55 µm spacing. Images were analyzed using the CellPathfinder software (version R3.04.02). Cells were detected and gated using a machine learning algorithm[28].

**Liability panel screening by DSF.** Recombinantly expressed proteins[32] (produced in-house, for constructs and sequences, refer to Supplementary Table 3) of the liability panel targets were diluted in a buffer containing 10 mM HEPES, pH 7.5 and 500 mM NaCl to a concentration of 2 µM. SYPRO Orange (Invitrogen, #S6650) was added as a 1:1000 dilution, the protein-dye mixtures were transferred to a 384 well plate, and the compounds were added using the Echo 550 liquid handler (Labcyte, San Jose, California, USA, #001-10080) to a final concentration of 20 µM. Temperature-dependent unfolding profiles were measured on a QuantStudio 5 real-time PCR machine (Thermo Fisher, Waltham, Massachusetts, USA). Excitation and emission filters were set to 465 nm and 590 nm, respectively. The temperature increase was set to 3 °C/min to a maximum temperature of 85 °C. Experiments were run as technical duplicates. Data were analyzed with

the Thermal Shift Software (version 1.4, Thermo Fisher) using the Boltzmann equation to determine the inflection point of the transition curve. Differences in melting temperature are given as ΔTm values in °C.

## CG compound profiling

**Gal4 hybrid reporter gene assays.** HEK293T cells were cultured in DMEM, high glucose, supplemented with 10% FBS, sodium pyruvate (1 mM), penicillin (100 U/mL), and streptomycin (100 µg/mL) at 37 °C and 5% $CO_2$ to a max. confluence of 70–80%, and seeded in clear 96-well plates (30,000 cells/well). After 20–24 h, the medium was changed to Opti-MEM without supplements, and the cells were transiently transfected using Lipofectamine LTX reagent (Invitrogen, Carlsbad, California, USA) according to the manufacturer's protocol with pFR-Luc (reporter gene; Stratagene, Agilent Technologies, Santa Clara, California, USA), pRL-SV40 (control gene for normalization of transfection efficiency and cell growth; Promega, Fitchburg, Wisconsin, USA) and one pFA-CMV-hNR-LBD clone coding for the hinge region and ligand binding domain (LBD) of a human NR of interest, or with pECE-SV40-Gal4-VP16[43] (gift from Lea Sistonen; Addgene plasmid #71728) as ligand-independent transcriptional inducer for control experiments. Five hours after transfection, the cells were incubated with the test compounds by changing the medium to Opti-MEM supplemented with penicillin (100 U/mL) and streptomycin (100 µg/mL) additionally containing 0.1% DMSO and the respective test compound or 0.1% DMSO alone as untreated control. After overnight (14–6 h) incubation, firefly and renilla luminescence were measured using the Dual-Glo® Luciferase Assay System (Promega) according to manufacturer's protocol on a Tecan Spark® 10 M multimode microplate reader (Tecan Group AG, Männedorf, Switzerland). Firefly luminescence was divided by renilla luminescence and multiplied by 1000 to obtain relative light units (RLU). Fold activation was obtained by dividing the mean RLU of the test compound at a respective concentration by the mean RLU of the untreated control. Experiments were performed in duplicates in at least three biologically independent repeats. For dose response curve fitting and calculation of $EC_{50}/IC_{50}$ values, the equation "[Agonist]/[Inhibitor] vs. response−Variable slope (four parameters)" was used in GrapPad Prism 9 (version 9.5.1, GraphPad software, La Jolla, CA, USA). The following pFA-CMV-hNR-LBD clones and reference ligands were used: THRα (pFA-CMV-hTHRα-LBD[56], 0.1 µM T3), THRβ (pFA-CMV-hTHRβ-LBD[56], 0.1 µM T3), RARα (pFA-CMV-hRARα-LBD[57], 1 µM retinoic acid), RARβ (pFA-CMV-hRARβ-LBD[57], 1 µM retinoic acid), RARγ (pFA-CMV-hRARγ-LBD[57], 1 µM retinoic acid), PPARα (pFA-CMV-hPPARα-LBD[58], 1 µM GW7647), PPARβ (pFA-CMV-hPPARδ-LBD[58], 1 µM L-165,041), PPARγ (pFA-CMV-hPPARγ-LBD[58], 1 µM rosiglitazone), RORα (pFA-CMV-hRORα-LBD[59], 1 µM T0901317), RORβ (pFA-CMV-hRORβ-LBD[59], 1 µM T0901317), RORγ (pFA-CMV-hRORγ-LBD[59], 1 µM cintirorgon), revERBα (pFA-CMV-hrevERBα-LBD[60], 10 µM SR9011), revERBβ (pFA-CMV-hrevERBβ-LBD[60], LXRα (pFA-CMV-hLXRα-LBD[61], 1 µM T0901317), LXRβ (pFA-CMV-hLXRβ-LBD[61], 1 µM T0901317), FXR (pFA-CMV-hFXR-LBD[62], 1 µM GW4064), VDR (pFA-CMV-hVDR-LBD[57], 1 µM calcitriol), PXR (pFA-CMV-hPXR-LBD[57], 1 µM SR12813), CAR (pFA-CMV-hCAR-LBD[57], 1 µM CITCO), HNF4α (pFA-CMV-hHNF4α-LBD[63], 10 µM compound 9 from [63]), RXRα (pFA-CMV-hRXRα-LBD[57], 1 µM bexarotene), TR2 (pFA-CMV-hTR2-LBD[64]), TLX (pFA-CMV-hTLX-LBD[65], 30 µM propranolol), Nur77 (pFA-CMV-hNur77-LBD[66], 100 µM amodiaquine), and SF1 (pFA-CMV-hSF1-LBD[67]).

## Assays for NR1 CG set application

**NFκB activity assay in astrocytes.** T98G cells (ATCC, CRL-1690) were cultured in DMEM high glucose, supplemented with 10% FBS, sodium pyruvate (1 mM), penicillin (100 U/mL) and streptomycin (100 µg/mL) at 37 °C and 5% $CO_2$. Necrotic soups were prepared by harvesting and resuspending T98G cells in PBS to a final concentration of 7 × 10$^6$ cells/mL and freezing (−80 °C) and thawing (37 °C) the suspension four

times, followed by centrifugation at 1000 g for 5 min. Aliquots were stored at −80 °C and centrifuged again at 13,300 g for 5 min right before use. T98G cells were seeded in 96-well plates (1 × 10⁴ cells per well). After 24 h, medium was changed to Opti-MEM™ without supplements and cells were transiently transfected with 4×NFκB Luc (gift from Johannes A. Schmid; Addgene plasmid #111216) and pRL-SV40 (Promega) using Lipofectamine 3000 (Invitrogen) transfection reagent according to the manufacturer's protocol. 5 h after transfection, the medium was changed to a mixture of Opti-MEM (60%) supplemented with penicillin (100 U/mL) and streptomycin (100 μg/mL), PBS (10%) and necrotic soup (30%) as well as the CG library compounds (at the recommended concentration) and 0.1% DMSO or 0.1% DMSO alone as negative control. Following incubation for 24 h, cells were assayed for firefly and renilla luciferase activity using the Dual-Glo Luciferase Assay System (Promega) according to the manufacturer's protocol and a Tecan Spark® 10 M multimode microplate reader (Tecan Group AG). Normalization for transfection efficiency and cell growth was achieved by division of firefly luciferase data by renilla luciferase data and multiplying the value by 1000, resulting in RLU. Relative NFκB activity was obtained by dividing the mean RLU of a test sample by the mean RLU of the DMSO control. Experiments were performed in four biologically independent repeats.

**Autophagy Flux Assay.** pMRX-IP-GFP-LC3-RFP-LC3ΔG was a gift from Noboru Mizushima (The University of Tokyo, Tokyo, Japan; Addgene plasmid #84572[25,50]). hTERT RPE-1 cells (provided by Andrew Holland, Johns Hopkins School of Medicine) were seeded in 384-well plates (2000 cells/well) in 50 μL DMEM, supplemented with 10% FBS, penicillin (100 U/mL) and streptomycin (100 μg/mL), and incubated for 24 h before treatment. 50 μL of media containing either 2x final (recommended) concentration of the CG set or control compounds (final conc.: 0.1% DMSO, 250 nM Torin-1, 200 ng/mL Bafilomycin A1) were added, and images in three channels (fluorescence of GFP, RFP, as well as phase) were taken every two hours over 72 h using an IncuCyte (Sartorius, Germany). General autophagy flux was calculated from the ratio of the total integrated fluorescence intensities of GFP/RFP. Cell confluence is represented as % of covered area by cells. Each data point represents the averaged ratio or confluence obtained from three individual wells of the plate. Values for each individual compound and time point were normalized to the respective value of the negative control DMSO by subtracting the averaged ratio from the corresponding average ratio of the averaged DMSO control of the same 384-well plate. Experiments were performed in three biologically independent repeats.

**Cancer cell proliferation assays.** T98G (ATCC, CRL-1690) and A549 (ATCC, ACC-107) cells were cultured in DMEM high glucose, supplemented with 10% FBS, sodium pyruvate (1 mM), penicillin (100 U/mL) and streptomycin (100 μg/mL) at 37 °C and 5% CO₂. HT-29 cells (ATCC, ACC-299) were cultured in McCoy's modified medium 5 A supplemented with 10% FBS, penicillin (100 U/mL) and streptomycin (100 μg/mL) at 37 °C and 5% CO₂. T98G, A549, and HT-29 cells were seeded in 96-well plates (5 × 10³ cells per well). After 24 h, the medium was changed to DMEM (A549, T98G) supplemented with 0.5% FBS, sodium pyruvate (1 mM), penicillin (100 U/mL) and streptomycin (100 μg/mL) or McCoy's modified medium 5 A (HT29) supplemented with 0.5% FBS, penicillin (100 U/mL) and streptomycin (100 μg/mL) additionally containing the CG library compounds (at the recommended concentration) and 0.1% DMSO or 0.1% DMSO alone as an untreated control. Bexarotene (10 μM) and flavopiridol (100 μM) served as positive controls for inhibition of cancer cell proliferation and induction of cell death. Immediately after stimulation (t = 0 h) and repeatedly thereafter (t = 24 h, 48 and 72 h), cell confluence was determined using a Tecan Spark Cyto microplate reader (Tecan Group AG). Changes in confluence were expressed by normalization to t = 0 h. After 72 h, cells

were incubated for 30 min with 4 μM Hoechst 33342 (Thermo Scientific, #62249) and 0.05X Biotium Live-or-Dye NucFix™ Red Staining (Biozol, Eching, Germany, #BOT-320101-T) to determine the total amount of cells and necrotic cells, respectively. Fluorescence imaging was conducted on a Tecan Spark Cyto microplate reader (Hoechst33342: Ex 381–400 nm, Em 414–450 nm; NucFix™ Red: Ex 543–566 nm, Em 580–611 nm). Experiments were performed in three biologically independent repeats.

### Reporting summary
Further information on research design is available in the Nature Portfolio Reporting Summary linked to this article.

## Data availability
All data supporting the results of this study are available in the Supplementary Information, Supplementary Data, at zenodo [https://doi.org/10.5281/zenodo.10474037; https://zenodo.org/records/10474037], and in BioImage Archive (accession codes: S-BIAD145; S-BIAD730; S-BIAD733). Source data are provided with this paper as a Source data file.

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

## Acknowledgements

The authors thank Silke Duensing-Kropp, Dr. Sandra Häberle, Dr. Georg Höfner, Dr. Vaclav Nemec, Dr. Jörg Pabel, and Dr. Sandra Röhm. This work has received funding from the Innovative Medicines Initiative 2 Joint Undertaking (JU) under grant agreement No. 875510. The JU receives support from the European Union's Horizon 2020 research and innovation program, EFPIA, Ontario Institute for Cancer Research, Royal Institution for the Advancement of Learning McGill University, Kungliga Tekniska Hoegskolan, and Diamond Light Source Limited. This research was co-funded by the European Union (ERC, NeuRoPROBE, 101040355; to D.M.). Views and opinions expressed are, however, those of the author(s) only and do not necessarily reflect those of the European Union or the European Research Council. Neither the European Union nor the granting authority can be held responsible for them. A.T. was supported by the SFB 1177 'Molecular and Functional Characterization of Selective Autophagy' (259130777). Further support was received from the German Research Foundation (DFG) – FUGG grant numbers 515275293 (cell sorter) and 512574446 (CQ1 spinning disc confocal).

## Author contributions

L.I., E.S., and D.M. evaluated CG compound candidates and assembled the NR1 CG set. L.I., E.S., R.B., L.B., A.M., and L.E. performed experiments. L.I., E.S., R.B., S.M., A.S., J.M., and D.M. analyzed data. S.M., S.K., and D.M. conceived the study. D.M. supervised the project. L.I., E.S., and D.M. prepared the figures and wrote the manuscript.

## Funding

## Competing interests

The authors declare no competing interests.
