## [Peer Review File · Nature Communications]

Chemogenomics for NR1 nuclear hormone receptorsREVIEWER COMMENTS

Reviewer #1 (Remarks to the Author):

They authors have systematically pieced together a focused chemogenetic library of ~70 compounds targeting the NR1 nuclear hormone receptor class. Others have assembled similar class libraries (i.e. Structural Genomics Consortium's kinase chemogenomics set (KCGS)) or broader collections such as Novartis' "Mechanism of Action Box" and Pfizer's chemogenomic library. Previous collections, such as Novartis collection, do outline a number of NR1 probes but this manuscript provides a much richer set of tool compounds (with diverse chemotypes) AND validation data. This NR1-enriched set of chemical probes is a valuable reference to enable better examination of nuclear hormone receptor biology which is a tough class of drug targets and will find strong use and adaptation of both academic and industry scientists conducting phenotypic screening or probing the connection of nuclear hormone systems and pathway biology.

One advancement the authors leverage in the manuscript is the aspect that their collection is 100% commercially available. This has a big advantage to enabling others to easily build and duplicate. They also do a nice job of validating the collection for cellular toxicity as well as on- and off-target activity across the NR1 family. This provides readers with a nice guidance for suggested starting concentrations to get the best potential results. While this is a good starting point for accessibility the authors should probably mention that by limiting their set to commercially available compounds, they are omitting several more powerful probes (particularly in the ROR and PPAR classes).

To better enable readers to benefit from the authors work it would be nice if they could include in their tables of their library 1) CAS# for each of the compounds so that readers could easily commercially source the compounds. 2) gene alias to the Gene names used as many readers are more likely to be familiar and use THR α (LXR, CAR, PPAR γ , ROR γ , etc) than NR1A1 (NR1A, NR1B, NR1C,...) 3) primary reference/citations of compounds to the associated target(s) annotated (this is partially done in the SI where they reference literature reported activities).

While the authors acknowledge that at points to expand the target class coverage and expand chemotype diversity they included a handful of compounds (i.e. amiodarone, meclizine, lovastatin,...) While they have nicely validated the activity of these compounds on NR1 family members they are not the primary targets of these compounds. Including them in cellular based screening may provide activities that will be misinterpreted as their NR1 activities rather than their primary targets. Because there is some redundancy, I'd strongly recommend removing the following 3 compounds from the set: 1) lovastatin - HMGCR; potentially misinterpreting the authors autophagy results 2) Meclizine - HRH1 & PCYT2 3) amiodarone - KCNH2 and CACNA2D2. At the very least these other targets should be annotated in the table. Amiodarone should especially be removed as in addition to its primary targets it is known to inhibit PLA2G15 which induces phospholipidosis (10.1016/j.jlr.2021.100089) and is known to be a lysosmotropic drug that can lead to a number of cellular pharmacological readouts (10.2174/1389200218666170925125940 & 10.1021/mp200641e & 10.1016/j.taap.2011.12.004) which could be mistaken for the NR1 activities. The use of 'bad' chemical probes can have long-standing negative effects across the scientific literature (Promise and Peril of chemical probes; 10.1038/nchembio.1867).

Its surprising and unclear why a few other compounds didn't make it into the collection. For example the compound HX-531 (CAS:188844-34-0; RXR antagonist) which has a different chemotype from the others included in the set was not included. This chemotype with HX-531 and Cmpd2170 (10.1038/s41586-020-2776-9 & 10.1016/j.bmcl.2007.06.080) are often more potent and selective tool compounds for the RXR class. It is perhaps a bit unclear how the authors arrived at the first list of 75 compounds and this could be better clarified perhaps by a pipeline/flowchart image in the supplementary figures.

I'd recommend publishing with minor revisions to help enhance the manuscript as a resource and

tool for others. As mentioned above adding a bit more reference info around the compounds (CAS, ref, gene alias, etc). I also strongly feel amiodarone should be excluded from the paper. There are a few minor grammatical and typos that should be fixed.

Reviewer #2 (Remarks to the Author):

The manuscript by Isigkeit, et al. curated a set of 68 compounds that target the NR1 family of nuclear receptors composed of 19 out of 48 NRs in humans. This set of compounds, if properly identified, may be useful for target identification and validation for the NR1 subset of NRs.

Targeted compound screening for some receptors in the NR1 family, specifically PPAR, FXR, LXR, and RAR have been extensively done in literature and is reviewed in detail by Ishigami-Yuasa and Kagechika (Int J Mol Sci, 2020). Moreover, a chemogenomic approach to drug discovery led to a highly curated list of 214 cardiovascular targets including nuclear receptors (Cases and Mestres, Drug Discovery Today, 2009). This present work by Isigkeit, et al. focuses on the NR1 family, which they found to have involvement with neuroinflammation, autophagy and cancer cell apoptosis using the curated set of 68 compounds. This is a significant proof-of-concept demonstration of chemogenomics, provided these compounds were properly identified.

However, after close examination of the data presented, I found major shortcomings and inconsistencies that need to be addressed:

- (1) Figure 1a does not show cytotoxicity/viability assay results (See 3rd sentence under "Activity profiling and CG compound selection", page 3).
- (2) Was the population doubling time of each cell line considered for the setup? 24 hours may not be enough to calculate the growth rate for some cell lines. The literature that was cited from Nature Methods, Hafner et al., (2016), performed the treatment course for 72 hours. (See 3rd sentence mentioned "different time points 6h, 12h, 18h and 24h" under "Activity profiling and CG compound selection", page 3).
- (3) In the last sentence on page 3, you mentioned "Eleven compounds (amiodarone, AHPN, SR9009, SR1078, SR2211, T0901317, GSK2033, DY268, doxercaliferol, lovastatin and meclizine) exhibiting a $GR \leq 0.5$ or inducing atypical cellular phenotypes (Supplementary Figure 1), were further evaluated in a high-content microscopy-based multiplex assay." It is unclear what you meant by atypical cellular phenotypes. Results in Supp Figure 1 need to be further discussed in the text.
- (4) One of your criteria for the CG compounds is the lack of cytotoxicity at the recommended concentration for cell-based screening (last paragraph of page 4). Nevertheless, "At the recommended concentration (1 μ M), DY268-treated HEK293T (>50%), MRC9 (>50%) and U-2 OS (slightly <50%) cells showed moderately decreased cell numbers supporting suitability of the CG compound candidate for inclusion in the set" (1st paragraph of page 4). Kindly explain why you decided to include DY268 in the CG set in spite of its substantial cytotoxicity at 1 μ M.
- (5) On page 3, it was mentioned that "NR1 CG compound candidates were extracted from this dataset from all annotated NR1 ligands (30862 compounds with potency $\leq 10 \mu$ M) based on i) community agreed criteria (Ref. 22) for cellular potency ($\leq 10 \mu$ M, preferably $\leq 1 \mu$ M)." Ref 22 is a link to EubOPEN homepage that does not explicitly mention these criteria. I recommend that you use journal article references instead.
- (6) On paragraph , page 4, it was mentioned that "compound induced increase in the protein melting temperature (ΔT_m) $> 2^\circ\text{C}$ was considered relevant, but the authors failed to explain how this cutoff was established. It seems that this was arbitrarily assigned.

(7) On page 5, what is the significance of Figure 2a? It was never discussed anywhere in the manuscript.

(8) In the Figure 2 caption, it was mentioned that "Proteins were used at 2 μ M; Staurosporine (ABL1, AURKA, CDK2, FGFR3 and GSK3B), (+)- JQ1 (BRD4), GSK6853 (BRPF1), PK016714a (CSNK1D), GDC-0994 (MAPK1) and IACS-9571 (TRIM24) served as references (20 μ M)." Aside from ABL1, AURKA and FGFR3 (whose inclusion were justified in the manuscript), why are the other aforementioned proteins selected for non-specific targeting? You mentioned in the text that these are "highly druggable proteins". What are the criteria you used for a protein to be considered highly druggable? Any references?

(9) In Figure 2b, last column is labeled "Reference". It is unclear what compound was used as "Reference" for the respective off-target proteins. I interpret this as positive control, but the identity of the positive control compound(s) is/are not identified.

(10) On page 7, Table 1: I can't see the literature sources of these "recommended concentrations." What specimen (cell, animal, protein, etc.) was used to establish those EC50 and IC50? The EC/IC50 for one specimen may be different from the cells that the authors are using. How did the authors arrive at those recommended concentrations?

(11) On Table 1, it's much simpler to just mention in column 4 that it is an agonist, inverse agonist or antagonist. The color coding scheme is unnecessary and confusing.

(12) What is the significance of Figure 4b on page 8? It was never discussed.

(13) On Figure 5, page 10, why did the authors exclude HEK293T, U2OS, and MRC-5 when they used it initially for the cytotoxicity assay?

(14) The authors used the nuclear receptor protein names on the phylogenetic tree in Figure 1a (page 2) but used "gene name format" (e.g., NR1A1) in Table 1 columns 2 and 5 on page 7. This is going to be confusing and inconvenient for readers as they have to use an online search engine (e.g., Google) to find the protein name of the NR. At the very least, translate the gene names to protein names in a legend under Table 1.

(15) You mentioned that "CG compounds' Tanimoto similarity (Ref 24) computed on Morgan fingerprints (Figure 4c) and scaffold analysis (Supplementary Figure 3) revealed only one scaffold contained in more than one ligand (retinoic acid (NR1B) and SR12813 (NR1I)) underlining high chemical diversity of all NR1 CG compounds." This needs to be further discussed as it is unclear how Figure 4c and Supp Figure 3 showed this finding. Figure 4c does not even have axis labels.

(16) In the sentence that follows, you mentioned "These compounds fully cover the chemical space of known NR1 modulators ($EC_{50}/IC_{50} \leq 100 \mu$ M) without clustering, as illustrated by t-SNE (Ref 45) (Figure 4d)." Briefly discuss what t-SNE is and explain how Figure 4d shows "no clustering" when obviously there are some data sets that seem to cluster (e.g., NR1C and NR1B have distinct clustering or groupings, similar to those in principal components analysis plots or PCA). Why did you use t-SNE instead of PCA?

(17) The authors concluded that "The final NR1 CG library comprising 68 deliberately selected compounds has been designed for use as a set enabling confident target identification and target deconvolution with several orthogonal chemical tools per NR1 protein." How can you justify confident target identification of NR1D (REVERB) if there is only a limited number of CG compounds (SR8278, SR9009, SR9011) identified? Are three compounds enough to differentiate REVERB α from REVERB β ? How many CG compounds are necessary to reach statistical significance for confident identification and deconvolution?

(18) The authors also concluded that "Additionally, lack of cytotoxicity and off-target activities in a panel of liability targets at the recommended concentration render this set as valuable and validated tool to explore uncharted biology of the NR1 family." This is not substantially supported by the data presented. For example, DY268 is cytotoxic but it was still included in the CG set.

Furthermore, most of the off target proteins (Figure 2b) selected were seemingly arbitrarily chosen and not justified in the manuscript. It is also a glaring oversight that the authors did not bother to test the curated CG compounds for non-specific targeting with the other members of the nuclear receptor superfamily (i.e, the other 29 nuclear receptors that are not members of the NR1 family). How can you be so sure these are NR1 specific if no validation was performed for nonspecific interaction with other non-NR1 nuclear receptors?

(19) The authors mentioned that "The CG compound candidates were acquired and analyzed for identity and purity ($\geq 95\%$) by LC-MS" on page 3. The LC-MS methodology was described in the supporting information. However, I cannot find the LC-MS data anywhere in the manuscript and supporting information in order to verify the identity and purity of these CG compounds. Furthermore, NMR should have been performed to validate correct stereochemistry of compounds with chiral centers and differentiate isobars with similar LC retention time.

(20) In Supplementary Table 1, column 3, the EC50/IC50 reported from literature do not have error bars. These error bars should be added to easily gauge the precision of these determinations. Moreover, for values determined by the authors in an assay, kindly report the results using correct significant figures. For example, 5.98 ± 4.39 μM should be reported as 6 ± 4 μM and 0.111 ± 0.036 μM should be 0.11 ± 0.04 μM . For column 4, just mention the type of agonism and remove the unnecessary color coding scheme.

(21) Supplementary references have to also appear in the main article references so that readers could conveniently cite them.

In aggregate, the conclusions of the authors are not substantially supported by data. Additional evidence are needed. There are some flaws in the methodology that need to be addressed. For example, why did the authors exclude HEK293T, U2OS, and MRC-5 in Figure 5 when they used it initially for the cytotoxicity assay? Most of the methodology appears to be sound but specific details such as cutoff criteria for the assays performed were not properly explained or established. Without proper cutoff criteria, how can we trust the curated CG compounds? Furthermore, the authors failed to demonstrate rigor by not showing any LC-MS data to establish the identity and purity of compounds. They also did not acquire NMR data to ensure that the stereochemistry of compounds with chiral centers are correct. The methods as described in the Supporting Information have sufficient details for the work to be reproduced.

In its current form, the manuscript is not suitable for publication in this journal.

Reviewer #3 (Remarks to the Author):

This manuscript reports a curated set of 68 ligands reported to modulate the transcriptional activity of the "NR1" subclass of nuclear receptor (NR) transcription factors (19 out of the 48 total human NRs). The curated ligand set was generated by testing compounds in several assays including cytotoxic effects in three general cell lines; and specificity via four different assays including binding to 10 non-NR proteins via (differential scanning fluorimetry assay), transcriptional activity in Gal4-NR LBD hybrid fusion luciferase reporter assay, transcriptional activity against RXRa that functions as a co-receptor for some but not all of the NR1s, and a Gal4-VP16 control assay. A feature analysis grouping ligands by NR1 sub-subclass (e.g., NR1A, NR1B,...,NR1I) and potency analysis is provided for the ligands set. Finally, the ligand set was used in a "chemogenomics" (CG) approach for identification of NR1s that influence various cellular phenotypic endpoints including NF κ B signaling that occurs in inflammation, autophagy, cell differentiation in a lung cancer cell model.

The main strength of this manuscript is the report of a curated set of commercially available NR1 ligands. The assays used to test the compounds have been used extensively in the NR field previously, and the data reported appear to be robust and rigorous.

The main weakness of this manuscript is the underlying significance of the work. The CG phenotypic profiling of the NR1 ligands are described with the underlying assumption that the findings are novel. However, a quick search of the literature for some of the NR1 proteins and their ligands shows that many (most? all?) of the findings here have been previously reported by others. In one sense, it may be considered positive that this current manuscript can replicate previously reported findings, addressing reproducibility. However, none of the previously reported studies are cited here and the authors do not delineate what is novel vs. previously reported and validated or refuted.

Other weaknesses:

The authors switch back and forth between various NR1 nomenclature (e.g. NR1A or THR). A few of these are called out in the manuscript text. However, it would help reader comprehension if both types of names were used in at least one figure (e.g., Figure 1a) and one table (e.g., Table 1).

Figure 1 describes the general mechanism of ligand-dependent NR transcriptional activation. However, this model does not account for some ligands in the curated set such as T091317 that functions as an agonist of NR1H/LXRs (likely via the model shown in Figure 1c) but an inverse agonist of NR1F/RORs where ligand binding would in principle decrease coactivator binding and increase corepressor binding. Furthermore, this model does not account for receptors in the NR1D sub-subfamily that interact exclusively with corepressor proteins because of their unique LBD structure that lacks the AF-2 "helix 12" important for interacting with coactivator proteins.

Most of the curated ligands are agonists of the NR1s. Non-agonist ligands are included for some NR1s, but there are other NR1s for which non-agonists (antagonists and inverse agonists) are available but not included. The authors could perform a more comprehensive search of non-agonists of other NR1s and add them to the curated ligand set and characterize them in the same activity assays. Otherwise, the rationale for including non-agonists for some NR1s but not others should be justified.

There are two recommendations for the data in Table 1. First, it would be useful if this information was also provided as a downloadable file (Excel or CSV) in the SI for readers to download. Second, the ligand "type" (agonist, inverse agonist, and antagonist) is a color coded scheme — it would be useful in a new column was added to this table where the words "agonist", "inverse agonist", and "antagonist" are used in particular if the SI Table file were made available so readers can search for those terms.

There is no general description of the various "Feature analyses" in Figure 4 to inform a broad audience (such as the readership of this journal) why this information is generally useful and what data were used in the analyses. Why do we care about subskeletons, Jaccard-Tanimoto similarity, t-SNE, etc.?

Raw data underlying all of the plots in the manuscript and SI document should be made available as an Excel or CSV file.

LUDWIG-
MAXIMILIANS-
UNIVERSITÄT
MÜNCHEN

DEPARTMENT OF PHARMACY
CHAIR OF PHARMACEUTICAL AND MEDICINAL CHEMISTRY
PROF. DR. DANIEL MERK

Prof. Dr. Daniel Merk
Chair of Pharmaceutical and
Medicinal Chemistry
Department of Pharmacy
Ludwig-Maximilians-Universität
Butenandtstr. 5-13,
D-81377 Munich
Germany
+49-89-2180-77249
daniel.merk@cup.lmu.de

Reg.: Revision of manuscript " Chemogenomics for NR1 nuclear hormone receptors"

München, 19.03.2024

Dear Reviewers

Thank you very much for performing peer-review of our manuscript on "Chemogenomics for NR1 nuclear hormone receptors", for your positive feedback and for your very constructive comments. Your input has helped improve the quality of the NR1 CG set and the manuscript. We have addressed all your concerns and suggestions in full. Our point-by-point answers are given below. We hope the revised manuscript meets your expectations and thank you for your further consideration.

Sincerely

Daniel Merk

REVIEWER COMMENTS

Reviewer #1 (Remarks to the Author):

They authors have systematically pieced together a focused chemogenetic library of ~70 compounds targeting the NR1 nuclear hormone receptor class. Others have assembled similar class libraries (i.e. Structural Genomics Consortium's kinase chemogenomics set (KCGS)) or broader collections such as Novartis' 'Mechanism of Action Box' and Pfizer's chemogenomic library. Previous collections, such as Novartis collection, do outline a number of NR1 probes but this manuscript provides a much richer set of tool compounds (with diverse chemotypes) AND validation data. This NR1-enriched set of chemical probes is a valuable reference to enable better examination of nuclear hormone receptor biology which is a tough class of drug targets and will find strong use and adaptation of both academic and industry scientists conducting phenotypic screening or probing the connection of nuclear hormone systems and pathway biology.

One advancement the authors leverage in the manuscript is the aspect that their collection is 100% commercially available. This has a big advantage to enabling others to easily build and duplicate. They also do a nice job of validating the collection for cellular toxicity as well as on- and off-target activity across the NR1 family. This provides readers with a nice guidance for suggested starting concentrations to get the best potential results. While this is a good starting point for accessibility the authors should probably mention that by limiting their set to commercially available compounds, they are omitting several more powerful probes (particularly in the ROR and PPAR classes).

We thank the Reviewer very much for evaluating our manuscript, for the positive feedback and for the constructive criticism. We are glad that the Reviewer sees the benefit of covering the NR1 family of nuclear hormone receptors with a rich set of extensively annotated tools that are all commercially available. We have addressed the Reviewer's comments and suggestions in full as outlined below. We hope the revised version of the manuscript and CG set meet the Reviewer's expectations.

We thank the Reviewer for the comment that the omission of some potentially better compounds from literature in favor of commercially available tools should be mentioned. As suggested, we have added a sentence to the candidate selection paragraph stating that commercial availability was prioritized and may have led to omission of potentially more potent alternative compounds.

To better enable readers to benefit from the authors work it would be nice if they could include in their tables of their library 1) CAS# for each of the compounds so that readers could easily commercially source the compounds. 2) gene alias to the Gene names used as many readers are more likely to be familiar and use THRα (LXR, CAR, PPAR α , ROR α , etc) than NR1A1 (NR1A, NR1B, NR1C,...) 3) primary reference/citations of compounds to the associated target(s) annotated (this is partially done in the SI where they reference literature reported activities).

Revised. We thank the Reviewer for this important comment. To improve the usability of the CG set by the community, we have added several items to the Supplementary Information:

- Supplementary Table 1 in the main SI pdf has been revised to additionally contain the CAS numbers of all compounds in the CG set and the relevant literature references for all CG compounds.

- We provide a computer-readable file (.xls) containing all compounds, their CAS numbers, their biological activities, and the DOIs for literature references containing bioactivity data. This file is also publicly available at [10.5281/zenodo.10474037](https://zenodo.org/record/10474037)

- We provide a human readable file (pdf) for each compound containing information on chemical structure, CAS number, biological activities, relevant literature references, and analytical data (NMR, LC-UV, LC-MS) for reference/comparison. These files are also publicly available at [10.5281/zenodo.10474037](https://zenodo.org/record/10474037)

We have also unified the use of gene names and NR nomenclature throughout the manuscript/figures and provide their correlation in Fig. 1a to help the reader understand both systems.

While the authors acknowledge that it points to expand the target class coverage and expand chemotype diversity they included a handful of compounds (i.e. amiodarone, meclizine, lovastatin,...) While they have nicely validated the activity of these compounds on NR1 family members they are not the primary targets of these compounds. Including them in cellular based screening may provide activities that will be misinterpreted as their NR1 activities rather than their primary targets. Because there is some redundancy, I'd strongly recommend removing the following 3 compounds from the set: 1) lovastatin - HMGCR; potentially misinterpreting the authors autophagy results 2) Meclizine - HRH1 & PCYT2 3) amiodarone - KCNH2 and CACNA2D2. At the very least these other targets should be annotated in the table. Amiodarone should especially be removed as in addition to its primary targets it is known to inhibit PLA2G15 which induces phospholipidosis (10.1016/j.jlr.2021.100089) and is known to be a lysosmotropic drug that can lead to a number of cellular pharmacological readouts (10.2174/1389200218666170925125940 & 10.1021/mp200641e & 10.1016/j.taap.2011.12.004) which could be mistaken for the NR1 activities. The use of 'bad' chemical probes can have long-standing negative effects across the scientific literature (Promise and Peril of chemical probes; 10.1038/nchembio.1867).

Revised. We thank the Reviewer very much for this very important and constructive comment. We have reconsidered the inclusion of the three compounds in question (amiodarone, meclizine, lovastatin) in the final CG set, sought for alternatives and performed additional profiling. We identified KB-130015 as a suitable replacement for amiodarone as THR antagonist, and famprofazone as well as oxatomide as suitable replacements for lovastatin as PXR agonist. Amiodarone and lovastatin have been removed from the set as suggested. In the case of meclizine, we decided to stick to our original selection and to keep the compound in the set. We have evaluated PK11195 as potential alternative for meclizine but found this compound to be highly unselective (modulation of multiple NR1 receptors) and to exhibit non-specific effects in the Gal4-VP16 control experiment. In contrast, all results from profiling of meclizine supported suitability for the application in CG for NR1 receptors.

It's surprising and unclear why a few other compounds didn't make it into the collection. For example the compound HX-531 (CAS:188844-34-0; RXR antagonist) which has a different chemotype from the others included in the set was not included. This chemotype with HX-531 and Cmpd2170 (10.1038/s41586-020-2776-9 & 10.1016/j.bmc.2007.06.080) are often more potent and selective tool compounds for the RXR class. It is perhaps a bit unclear how the authors arrived at the first list of 75 compounds and this could be better clarified perhaps by a pipeline/flowchart image in the supplementary figures.

We thank the Reviewer very much for this valuable suggestion. We fully agree that these compounds are very good candidate compounds for chemogenomics but their target RXR (NR2B) does not belong to the NR1 family, so they are out of the scope of this study. We are working on a second set covering the remaining nuclear hormone receptors and have included HX-531 and Cmpd2170 as candidates for this set.

I'd recommend publishing with minor revisions to help enhance the manuscript as a resource and tool for others. As mentioned above adding a bit more reference info around the compounds (CAS, ref, gene alias, etc). I also strongly feel amiodarone should be excluded from the paper. There are a few minor grammatical and typos that should be fixed.

We thank the Reviewer again for the evaluation of our study and the very constructive comments. We feel that we have fully addressed the Reviewer's concerns and suggestions and that the study has strongly benefitted from this. We have also performed proof-reading of the manuscript to remove typos and grammatical errors. We hope the revised manuscript meets the Reviewer's expectations.

Reviewer #2 (Remarks to the Author):

The manuscript by Isigkeit, et al. curated a set of 68 compounds that target the NR1 family of nuclear receptors composed of 19 out of 48 NRs in humans. This set of compounds, if properly identified, may be useful for target identification and validation for the NR1 subset of NRs.

Targeted compound screening for some receptors in the NR1 family, specifically PPAR, FXR, LXR, and RAR have been extensively done in literature and is reviewed in detail by Ishigami-Yuasa and Kagechika (Int J Mol Sci, 2020). Moreover, a chemogenomic approach to drug discovery led to a highly curated list of 214 cardiovascular targets including nuclear receptors (Cases and Mestres, Drug Discovery Today, 2009). This present work by Isigkeit, et al. focuses on the NR1 family, which they found to have involvement with neuroinflammation, autophagy and cancer cell apoptosis using the curated set of 68 compounds. This is a significant proof-of-concept demonstration of chemogenomics, provided these compounds were properly identified.

We thank the Reviewer very much for performing peer-review of our manuscript, for the positive feedback, and for the very constructive suggestions to improve the selection and profiling of the NR1 CG set. We have addressed all the Reviewer's comments and concerns; point-by-point answers are given below. The Reviewer's input has strengthened the NR1 CG set and improved the quality of our study. We are very grateful for this constructive critique.

However, after close examination of the data presented, I found major shortcomings and inconsistencies that need to be addressed:

(1) Figure 1a does not show cytotoxicity/viability assay results (See 3rd sentence under “Activity profiling and CG compound selection”, page 3).

Revised. We thank the Reviewer very much for spotting this error. This should read Figure 2a, which we have amended in the revised manuscript. We apologize for the mistake.

(2) Was the population doubling time of each cell line considered for the setup? 24 hours may not be enough to calculate the growth rate for some cell lines. The literature that was cited from Nature Methods, Hafner et al., (2016), performed the treatment course for 72 hours. (See 3rd sentence mentioned “different time points 6h, 12h, 18h and 24h” under “Activity profiling and CG compound selection”, page 3).

Revised. We thank the Reviewer for these remarks on the toxicity screening. We agree that 24 h may not be enough for a precise calculation of the growth rate. However, the purpose of this assay was not to determine the precise growth rate of the cell lines, but to provide a first filter of CG compounds and identify potentially toxic compounds for evaluation in the more sophisticated multiplex toxicity assay. We used model cell lines (HEK293T, U2OS and MRC-9) that are also often used in cellular assays and that exhibit quite different behavior (slow division time for non-transformed fibroblasts versus faster division time for HEK293T and U2OS). By observing the growth rate, which takes into account the division time of the cells, we compare the compounds' effects on three cells lines to reveal compounds requiring further evaluation. We have revised the manuscript text to better reflect the purpose of the first compound assessment.

(3) In the last sentence on page 3, you mentioned “Eleven compounds (amiodarone, AHPN, SR9009, SR1078, SR2211, T0901317, GSK2033, DY268, doxercalciferol, lovastatin and meclizine) exhibiting a $GR \leq 0.5$ or inducing atypical cellular phenotypes (Supplementary Figure 1), were further evaluated in a high-content microscopy-based multiplex assay.” It is unclear what you meant by atypical cellular phenotypes. Results in Supp Figure 1 need to be further discussed in the text.

Revised. As mentioned above, the primary cell viability assay was used as a first step to triage the compounds to be used in the secondary multiplex assay. We therefore included not only compounds that affected the growth rate, but also compounds that affected the phenotypic appearance of cells upon visual inspection. The mentioning of Suppl. Figure 1 at the bottom of page 3 is misleading as Suppl. Figure 1 only refers to the multiplex assay. We have now aligned the results described in the main text with Suppl. Figure 1.

(4) One of your criteria for the CG compounds is the lack of cytotoxicity at the recommended concentration for cell-based screening (last paragraph of page 4). Nevertheless, “At the recommended concentration (1 μ M), DY268-treated HEK293T (>50%), MRC9 (>50%) and U-2 OS (slightly <50%) cells showed moderately decreased cell numbers supporting suitability of the CG compound candidate for inclusion in the set” (1st paragraph of page 4). Kindly explain why you decided to include DY268 in the CG set in spite of its substantial cytotoxicity at 1 μ M.

Revised. We thank the Reviewer for raising this important point and noting that the reason for inclusion of DY268 were not clear enough. DY268 indeed exhibited slight toxicity which appeared to be mediated by tubulin effects according to the multiplex results. However, this effect was not observed in all cell lines and was weak at the recommended 1 μ M concentrations. This weak effect was considered acceptable especially since DY268 contributes importantly to the set as FXR antagonist for which there are no convincing commercially available alternatives. We have added these considerations and a remark, that moderate toxicity of DY268 should be taken into account when analyzing CG results, to the CG compound selection section.

(5) On page 3, it was mentioned that “NR1 CG compound candidates were extracted from this dataset from all annotated NR1 ligands (30862 compounds with potency \leq 10 μ M) based on i) community agreed criteria (Ref. 22) for cellular potency (\leq 10 μ M, preferably \leq 1 μ M).” Ref 22 is a link to EubOPEN homepage that does not explicitly mention these criteria. I recommend that you use journal article references instead.

Revised. We thank the Reviewer for this remark. We have added a literature reference as suggested.

(6) On paragraph , page 4, it was mentioned that “compound induced increase in the protein melting temperature (ΔT_m) > 2°C was considered relevant, but the authors failed to explain how this cutoff was established. It seems that this was arbitrarily assigned.

Revised. We thank the Reviewer very much for this important comment. The cutoff/threshold we used to evaluate the DSF liability panel assay was based on the overall variance of the negative control (DMSO) in the assay. We considered a thermal shift > 2xSD of the assay corresponding to 1.8 °C relevant. Originally, we rounded this value to 2 °C for simplification but have now revised the manuscript to present explanation of the threshold. The slightly lower cutoff did not change the results.

(7) On page 5, what is the significance of Figure 2a? It was never discussed anywhere in the manuscript.

Revised. We thank the Reviewer for spotting this error. Figure 2a describes the growth rate assessment and is discussed in the compound selection section, but mistakenly was labelled Figure 1a in the text (page 3). We apologize for the mistake and have corrected the wrong labeling.

(8) In the Figure 2 caption, it was mentioned that “Proteins were used at 2 μ M; Staurosporine (ABL1, AURKA, CDK2, FGFR3 and GSK3B), (+)- JQ1 (BRD4), GSK6853 (BRPF1), PK016714a (CSNK1D), GDC-0994 (MAPK1) and IACS-9571 (TRIM24) served as references (20 μ M).” Aside from ABL1, AURKA and FGFR3 (whose inclusion were justified in the manuscript), why are the other aforementioned proteins selected for non-specific targeting? You mentioned in the text that these are “highly druggable proteins”. What are the criteria you used for a protein to be considered highly druggable? Any references?

Revised. We thank the Reviewer for this comment. The explanation of the liability panel was brief in the first version to keep the overall manuscript short and avoid a long excursion on this topic. Therefore, we had focused on explaining the inclusion of three representative examples. However, we agree with the Reviewer that the target selection for the panel requires some more detail. We have revised the manuscript to motivate the inclusion of all targets in the liability panel and added references.

(9) In Figure 2b, last column is labeled “Reference”. It is unclear what compound was used as “Reference” for the respective off-target proteins. I interpret this as positive control, but the identity of the positive control compound(s) is/are not identified.

Revised. We thank the Reviewer for this remark. The last column of the heatmap for the liability panel screening refers to the positive control. We have updated the label to "pos. ctrl" to clarify this.

(10) On page 7, Table 1: I can't see the literature sources of these "recommended concentrations." What specimen (cell, animal, protein, etc.) was used to establish those EC50 and IC50? The EC/IC50 for one specimen may be different from the cells that the authors are using. How did the authors arrive at those recommended concentrations?

Revised. We thank the Reviewer very much for raising this point and for noting that the recommended concentrations were not clear enough. We have identified suitable concentrations for CG application based on the results of the potency, selectivity and toxicity profiling of the compounds. We have added this missing information to the CG compound selection section.

(11) On Table 1, it's much simpler to just mention in column 4 that it is an agonist, inverse agonist or antagonist. The color coding scheme is unnecessary and confusing.

Revised. We thank the Reviewer for this comment. Table 1 was updated as suggested.

(12) What is the significance of Figure 4b on page 8? It was never discussed.

Revised. We thank the Reviewer for spotting that Figure 4b was not mentioned in the text. It illustrates the potency distribution of the compounds selected for the final CG set. We have added the missing reference to Figure 4b in the text.

(13) On Figure 5, page 10, why did the authors exclude HEK293T, U2OS, and MRC-5 when they used it initially for the cytotoxicity assay?

We thank the Reviewer for this comment. HEK293T, U2OS and MRC-5 cells were deliberately chosen for the initial toxicity assay as they are not of cancer origin and often used (especially HEK293T) for cellular assays. The toxicity assay in these cell lines shall reveal non-specific toxicity of the candidate compounds which would prevent their use in the CG set. A549, T98G and HT29, in contrast, are cancer cell lines and the evaluation of cell-death induction in these cells by the CG set aimed to reveal specific, target-mediated toxicity by the CG compounds (for which non-specific toxicity had been excluded by the initial toxicity assay).

(14) The authors used the nuclear receptor protein names on the phylogenetic tree in Figure 1a (page 2) but used “gene name format” (e.g., NR1A1) in Table 1 columns 2 and 5 on page 7. This is going to be confusing and inconvenient for readers as they have to use an online search engine (e.g., Google) to find the protein name of the NR. At the very least, translate the gene names to protein names in a legend under Table 1.

Revised. We thank the Reviewer for this remark. We have unified the use of gene names and NR nomenclature throughout the manuscript and figures. Figure 1a now contains both types of nomenclature for reference.

(15) You mentioned that “CG compounds' Tanimoto similarity (Ref 24) computed on Morgan fingerprints (Figure 4c) and scaffold analysis (Supplementary Figure 3) revealed only one scaffold contained in more than one ligand (retinoic acid (NR1B) and SR12813 (NR1I)) underlining high chemical diversity of all NR1 CG compounds.” This needs to be further discussed as it is unclear how Figure 4c and Supp Figure 3 showed this finding. Figure 4c does not even have axis labels.

(16) In the sentence that follows, you mentioned “These compounds fully cover the chemical space of known NR1 modulators ($EC_{50}/IC_{50} \leq 100 \mu M$) without clustering, as illustrated by t-SNE (Ref 45) (Figure 4d).” Briefly discuss what t-SNE is and explain how Figure 4d shows “no clustering” when obviously there are some data sets that seem to cluster (e.g., NR1C and NR1B have distinct clustering or groupings, similar to those in principal components analysis plots or PCA). Why did you use t-SNE instead of PCA?

Revised. We thank the Reviewer very much for these remarks and for noting that the computational analysis of chemical diversity was not clear enough. We have revised and extended the discussion of this evaluation to better explain the results and their relevance. To improve readability of the heatmap in Figure 4c, we have added labels. We have also updated the explanation and discussion of the t-SNE and why we have not used PCA.

(17) The authors concluded that “The final NR1 CG library comprising 68 deliberately selected compounds has been designed for use as a set enabling confident target identification and target deconvolution with several orthogonal chemical tools per NR1 protein.” How can you justify confident target identification of NR1D (REVERB) if there is only a limited number of CG compounds (SR8278, SR9009, SR9011) identified? Are three compounds enough to differentiate REVERB α from REVERB β ? How many CG compounds are necessary to reach statistical significance for confident identification and deconvolution?

We thank the Reviewer very much for this remark. Although the number of suitable ligands for CG was lowest for NR1D as noted by the Reviewer, the three selected compounds still form a very valuable set as they are well annotated and, more importantly, comprise two agonists and one inverse agonist. As observed in the experiment to detect effects on neuroinflammation (Fig. 5a), this composition has a valuable impact as the agonists and the inverse agonist have opposite phenotypic outcomes. Despite only containing three NR1D modulators, the CG set therefore enables the discovery of NR1D mediated effects as intended. Regarding the two REVERB subtypes the Reviewer is perfectly right, that no discrimination is possible with any available ligand. This problem also occurs for many other NR like RXR (NR2B) which is much better studied but lack subtype-selective ligands. We have revised the section on the characteristics and chemical features of the NR1 CG set to better reflect these considerations.

(18) The authors also concluded that “Additionally, lack of cytotoxicity and off-target activities in a panel of liability targets at the recommended concentration render this set as valuable and validated tool to explore uncharted biology of the NR1 family.” This is not substantially supported by the data presented. For example, DY268 is cytotoxic but it was still included in the CG set. Furthermore, most of the off target proteins (Figure 2b) selected were seemingly arbitrarily chosen and not justified in the manuscript. It is also a glaring oversight that the authors did not bother to test the curated CG compounds for non-specific targeting with the other members of the nuclear receptor superfamily (i.e., the other 29 nuclear receptors that are not members of the NR1 family). How can you be so sure these are NR1 specific if no validation was performed for nonspecific interaction with other non-NR1 nuclear receptors?

Revised. We thank the Reviewer for these important comments.

- Regarding DY268: This compound had a slight anti-proliferative effect in the initial toxicity screening at 10 μM but not at 1 μM which is the recommended concentration for CG application. Follow-up evaluation in the more sophisticated multiplex toxicity assay showed no phenotypic effects of DY268. Also the off-target interactions of DY268 were only observed at 20 μM but not at the recommended concentration and the compound revealed favorable selectivity within the NR1 family as well as no non-specific effects on Gal4-VP16. These results fully supported suitability of DY268 for the CG set.

- The missing information on the selection of off-targets for the liability panel has been added (see comment 8)

- We agree with the Reviewer that selectivity data on NR outside the NR1 family are relevant in this context. We have profiled the entire CG set for modulation of representative members of the NR2, NR4 and NR5 family. The results are presented in Figure 2f.

We have clarified these points in the manuscript.

(19) The authors mentioned that “The CG compound candidates were acquired and analyzed for identity and purity ($\geq 95\%$) by LC-MS” on page 3. The LC-MS methodology was described in the supporting information. However, I cannot find the LC-MS data anywhere in the manuscript and supporting information in order to verify the identity and purity of these CG compounds. Furthermore, NMR should have been performed to validate correct stereochemistry of compounds with chiral centers and differentiate isobars with similar LC retention time.

Revised. We thank the Reviewer for raising this point and we agree that identity and purity data are important for the readers and users of the set as reference. We have measured NMR for all compounds to validate the correct identity. We have added all analytical data to the Supplementary Information in the form of one pdf file per compound containing structural information, biological activity data, relevant literature references, and all analytical data. These files are also publicly available at [10.5281/zenodo.10474037](https://doi.org/10.5281/zenodo.10474037)

(20) In Supplementary Table 1, column 3, the EC50/IC50 reported from literature do not have error bars. These error bars should be added to easily gauge the precision of these determinations. Moreover, for values determined by the authors in an assay, kindly report the results using correct significant figures. For example, 5.98+/-4.39 uM should be reported as 6+/-4 uM and 0.111+/-0.036 uM should be 0.11+/-0.04 uM. For column 4, just mention the type of agonism and remove the unnecessary color coding scheme.

Revised. We thank the Reviewer for this important remark. We have included the errors from literature as far as errors have been reported in the original publications and corrected the presentation of our in-house results to have the correct significant figures. The color-coding scheme in column 4 has been replaced by the labels agonist/antagonist/inverse agonist.

(21) Supplementary references have to also appear in the main article references so that readers could conveniently cite them.

We thank the Reviewer for this suggestion and agree that this would be reasonable to include the original references for all compounds. However, this would result in >150 references in total. We will consult with the editor on this matter.

In aggregate, the conclusions of the authors are not substantially supported by data. Additional evidence are needed. There are some flaws in the methodology that need to be addressed. For example, why did the authors exclude HEK293T, U2OS, and MRC-5 in Figure 5 when they used it initially for the cytotoxicity assay? Most of the methodology appears to be sound but specific details such as cutoff criteria for the assays performed were not properly explained or established. Without proper cutoff criteria, how can we trust the curated CG compounds? Furthermore, the authors failed to demonstrate rigor by not showing any LC-MS data to establish the identity and purity of compounds. They also did not acquire NMR data to ensure that the stereochemistry of compounds with chiral centers are correct. The methods as described in the Supporting Information have sufficient details for the work to be reproduced. In its current form, the manuscript is not suitable for publication in this journal.

We thank the Reviewer again for all the constructive suggestions that have clearly improved the quality of the NR1 CG set and its annotation. We have revised the manuscript and the CG set in line with the Reviewer's comments and feel that we have fully addressed all concerns.

Reviewer #3 (Remarks to the Author):

This manuscript reports a curated set of 68 ligands reported to modulate the transcriptional activity of the “NR1” subclass of nuclear receptor (NR) transcription factors (19 out of the 48 total human NRs). The curated ligand set was generated by testing compounds in several assays including cytotoxic effects in three general cell lines; and specificity via four different assays including binding to 10 non-NR proteins via (differential scanning fluorimetry assay), transcriptional activity in Gal4-NR LBD hybrid fusion luciferase reporter assay, transcriptional activity against RXR α that functions as a co-receptor for some but not all of the NR1s, and a Gal4-VP16 control assay. A feature analysis grouping ligands by NR1 sub-subclass (e.g., NR1A, NR1B,...,NR1I) and potency analysis is provided for the ligands set. Finally, the ligand set was used in a “chemogenomics” (CG) approach for identification of NR1s that influence various cellular phenotypic endpoints including NF κ B signaling that occurs in inflammation, autophagy, cell differentiation in a lung cancer cell model.

The main strength of this manuscript is the report of a curated set of commercially available NR1 ligands. The assays used to test the compounds have been used extensively in the NR field previously, and the data reported appear to be robust and rigorous.

The main weakness of this manuscript is the underlying significance of the work. The CG phenotypic profiling of the NR1 ligands are described with the underlying assumption that the findings are novel. However, a quick search of the literature for some of the NR1 proteins and their ligands shows that many (most? all?) of the findings here have been previously reported by others. In one sense, it may be considered positive that this current manuscript can replicate previously reported findings, addressing reproducibility. However, none of the previously reported studies are cited here and the authors do not delineate what is novel vs. previously reported and validated or refuted.

We thank the Reviewer very much for evaluating our manuscript and for the positive and constructive feedback. The Reviewer's input was very valuable to improve the manuscript. We have addressed all comments; point-by-point answers are given below.

We are very grateful for the Reviewer's remark that the novelty and the purpose of this study was not clear enough. The Reviewer is perfectly right, that the CG compounds selected for the set developed and characterized here have been reported previously. In the present study, we aimed to develop a highly annotated set of chemical tools for the NR1 family of nuclear receptors from the available modulators. Commercial availability of these compounds was a key aspect to make the set accessible to a broad community. Therefore, previously published tools have been prioritized. In validating the CG compound candidates, we identified several examples that either failed to exhibit the activity reported in literature or displayed non-specific effects or poor selectivity. All these results validating or refuting NR1 modulators are provided in the Supplementary Information (as human readable and computer readable files) together with the corresponding references to the original reports.

The final CG set is composed only of highly validated tools and has been optimized for chemical diversity. To our knowledge, this set is the first example of a highly optimized CG library covering an entire protein family underscoring the novelty of this study. It will be a very useful tool and resource for research on NR1 receptors. As all contained compounds are commercially available, the set is sustainable and can be used by anyone. It can also be obtained from us (EUbOPEN) on request. Additionally, our study may serve as a blueprint for the development of CG sets for other protein families. Lastly, our study also provides three exemplary applications of the NR1 CG set that have revealed new pharmacologies for further exploration and show the great potential of a highly curated and optimized CG library.

We have revised the manuscript in several sections to better reflect these considerations.

Other weaknesses:

The authors switch back and forth between various NR1 nomenclature (e.g. NR1A or THR). A few of these are called out in the manuscript text. However, it would help reader comprehension if both types of names were used in at least one figure (e.g., Figure 1a) and one table (e.g., Table 1).

Revised. We thank the Reviewer for this comment. As suggested, we have added both gene names and NR nomenclature to Figure 1a for reference and we have unified the nomenclature throughout the manuscript and Figures.

Figure 1 describes the general mechanism of ligand-dependent NR transcriptional activation. However, this model does not account for some ligands in the curated set such as T091317 that functions as an agonist of NR1H/LXR α s (likely via the model shown in Figure 1c) but an inverse agonist of NR1F/ROR α s where ligand binding would in principle decrease coactivator binding and increase corepressor binding. Furthermore, this model does not account for receptors in the NR1D sub-subfamily that interact exclusively with corepressor proteins because of their unique LBD structure that lacks the AF-2 “helix 12” important for interacting with coactivator proteins.

Revised. We thank the Reviewer for this important remark. We have extended Figure 1 by further subfigures to additionally illustrate the different types of NR ligands with their effects on the conformation (d) and the structural differences for NR1D receptors (e). We have added recent references to reviews on structural/molecular mechanisms of NR modulation to the caption of Figure 1d for further reading.

Most of the curated ligands are agonists of the NR1s. Non-agonist ligands are included for some NR1s, but there are other NR1s for which non-agonists (antagonists and inverse agonists) are available but not included. The authors could perform a more comprehensive search of non-agonists of other NR1s and add them to the curated ligand set and characterize them in the same activity assays. Otherwise, the rationale for including non-agonists for some NR1s but not others should be justified.

Revised. We thank the Reviewer very much for this important comment. We have indeed prioritized agonists for most NR1 receptors as they have no relevant constitutive activity but have complemented the agonists with one antagonist per subfamily. For NR1F (ROR α s), which exhibit high constitutive activity, we have prioritized inverse agonists and complemented them likewise with a few agonists. We think that the concept of prioritizing like this is reasonable and that the fewer antagonists are sufficient to orthogonally complement the agonists (likewise for inverse NR1F agonists). We have added these considerations to the manuscript text.

There are two recommendations for the data in Table 1. First, it would be useful if this information was also provided as a downloadable file (Excel or CSV) in the SI for readers to download. Second, the ligand “type” (agonist, inverse agonist, and antagonist) is a color coded scheme — it would be useful in a new column was added to this table where the words “agonist”, “inverse agonist”, and “antagonist” are used in particular if the SI Table file were made available so readers can search for those terms.

Revised. We thank the Reviewer very much for this remark. As suggested, Table 1 in the manuscript was updated to contain the terms agonist/antagonist/inverse agonist instead of the color code. Additionally, we provide an xls-file containing all compounds, their targets, their biological activity and relevant references as Supplementary Information.

There is no general description of the various “Feature analyses” in Figure 4 to inform a broad audience (such as the readership of this journal) why this information is generally useful and what data were used in the analyses. Why do we care about subskeletons, Jaccard-Tanimoto similarity, t-SNE, etc.?

Revised. We thank the Reviewer very much for noting that the importance of chemical diversity in the CG set was not clear enough to the broad audience. We have expanded the discussion of the feature analysis to (i) better explain the different analyses, and (ii) discuss their importance.

Raw data underlying all of the plots in the manuscript and SI document should be made available as an Excel or CSV file.

Revised. We thank the Reviewer for raising this point. We provide the raw data for all experiments as Supplementary Information files (xls) as suggested.

REVIEWERS' COMMENTS

Reviewer #1 (Remarks to the Author):

The authors have pieced together a nice chemogenetic library of chemical tool compounds targeting the NR1 family. After revision they have strengthened the library content, removing some of the more promiscuous compounds. The group has done a nice job of verifying the appropriate activities and annotating potential off-target liabilities. After assembling and characterizing the chemical collection they demonstrate a use in screening. They have also added additional information to enable others to reference and readily build their own collection from commercially available tool compounds.

The authors have addressed all of my concerns and is suitable for publication.

Reviewer #2 (Remarks to the Author):

Most of my critiques were satisfactorily addressed by the authors. However, I noticed that one of my comments (#15) was overlooked and the authors did not respond to my questions/concerns in the rebuttal page.

(15) You mentioned that "CG compounds' Tanimoto similarity (Ref 24) computed on Morgan fingerprints (Figure 4c) and scaffold analysis (Supplementary Figure 3) revealed only one scaffold contained in more than one ligand (retinoic acid (NR1B) and SR12813 (NR1I)) underlining high chemical diversity of all NR1 CG compounds." This needs to be further discussed as it is unclear how Figure 4c and Supp Figure 3 showed this finding. Figure 4c does not even have axis labels.

Nevertheless, upon checking the revised manuscript, Figure 4c was substantially improved and a detailed discussion was added in p. 9.

I believe the thoughtful and meticulous revision of the manuscript warrants its publication in the journal.

Reviewer #3 (Remarks to the Author):

The revised manuscript addresses most of my previous comments and concerns. There are a few commercially available compounds that I think should have been included in this ligand set that are not currently represented in the pharmacological classes including one for PPARalpha (GW6471 antagonist/inverse agonist that represses transcription) and two for PPARgamma (T0070907, an inverse agonist that represses transcription; and GW9662, a transcriptionally neutral antagonist). However, the ligand set reported by the authors here is still quite extensive. Another point is that Table 1 the new updated Supplementary Table 1 contains IC50/EC50 values, which reports on compound potency; however, no information is given on the relative efficacy of each compound (e.g., how much the compound changes transcription) relative to DMSO control (e.g., partial agonism vs. full agonism). I think this is an important feature currently missing that the authors could likely add relatively easily since they should have all of the luciferase reporter data from their potency analysis and appear to report these values in the new source data set files.

Author's Response to Reviewers:

Reviewer 2:

Most of my critiques were satisfactorily addressed by the authors. However, I noticed that one of my comments (#15) was overlooked and the authors did not respond to my questions/concerns in the rebuttal page. (15) You mentioned that “CG compounds' Tanimoto similarity (Ref 24) computed on Morgan fingerprints (Figure 4c) and scaffold analysis (Supplementary Figure 3) revealed only one scaffold contained in more than one ligand (retinoic acid (NR1B) and SR12813 (NR1I)) underlining high chemical diversity of all NR1 CG compounds.” This needs to be further discussed as it is unclear how Figure 4c and Supp Figure 3 showed this finding. Figure 4c does not even have axis labels. Nevertheless, upon checking the revised manuscript, Figure 4c was substantially improved and a detailed discussion was added in p. 9. I believe the thoughtful and meticulous revision of the manuscript warrants its publication in the journal.

Comments (15) and (16) of Reviewer 2 in the previous revision were related and we have therefore addressed them together. Our answer to comment (16) in the rebuttal also covers comment (15) and we have addressed the Reviewer's point in the previous revision. We apologize that this was not clear. However, Reviewer 2 realized that their concern had been addressed despite our missing answer. We thank the Reviewer again for the constructive input! (Regarding Figure 4c: it would make no sense to provide full axis labels (i.e., the compound names) because they would not be readable. The compounds were plotted in the order shown in Table 1 and this should be clear now)

Reviewer 3:

The revised manuscript addresses most of my previous comments and concerns. There are a few commercially available compounds that I think should have been included in this ligand set that are not currently represented in the pharmacological classes including one for PPARalpha (GW6471 antagonist/inverse agonist that represses transcription) and two for PPARg (T0070907, an inverse agonist that represses transcription; and GW9662, a transcriptionally neutral antagonist)). However, the ligand set reported by the authors here is still quite extensive. Another point is that Table 1 the new updated Supplementary Table 1 contains IC50/EC50 values, which reports on compound potency; however, no information is given on the relative efficacy of each compound (e.g., how much the compound changes transcription) relative to DMSO control (e.g., partial agonism vs. full agonism). I think this is an important feature currently missing that the authors could likely add relatively easily since they should have all of the luciferase reporter data from their potency analysis and appear to report these values in the new source data set files.

We thank the Reviewer very much for the positive feedback on the revised version. As further suggested, we have added information on efficacy of the compounds to Supplementary Table 1 and to the computer-readable full data table of the CG set in the Source Data file. Therein, we provide data from literature as far as available and our own data on the compounds' efficacy at the recommended concentration. We agree with the Reviewer that this information is very valuable to the reader/user, and we thank the Reviewer for this constructive comment!